# Decreasing uncertainty in flood frequency analyses by including historic flood events in an efficient bootstrap approach

Anouk Bomers[1], Ralph M. J. Schielen[1,2], and Suzanne J. M. H. Hulscher[1]

[1]University of Twente, Dienstweg 1, Enschede, The Netherlands
[2]Ministry of Infrastructure and Water Management-Rijkswaterstaat, Arnhem, The Netherlands

**Correspondence:** A. Bomers (a.bomers@utwente.nl)

**Abstract.** Flood frequency curves are usually highly uncertain since they are based on short data sets of measured discharges or weather conditions. To decrease the confidence intervals, an efficient bootstrap method is developed in this study. The Rhine river delta is considered as a case study. We use a hydraulic model to normalize historic flood events for anthropogenic and natural changes in the river system. As a result, the data set of measured discharges could be extended with approximately 600 years. The study shows that flood events decrease the confidence interval of the flood frequency curve significantly, specifically in the range of large floods. This even applies if the maximum discharges of these historic flood events are highly uncertain themselves.

## 1 Introduction

Floods are one of the main natural hazards to cause large economic damage and human casualties worldwide as a result of serious inundations with disastrous effects. Design discharges associated with a specific return period are used to construct flood defences to protect the hinterland from severe floods. These design discharges are commonly determined with the use of a flood frequency analysis (FFA). The basic principle of an FFA starts with selecting the annual maximum discharges of the measured data set, or peak values that exceed a certain threshold (Schendel and Thongwichian, 2017). These maximum or peak values are then used to identify the parameters of a probability distribution. From this fitted distribution, discharges corresponding to any return period can be derived.

Return periods of design discharges are commonly in the order of 500 years or even more, while discharge measurements have been performed only for the last 50-100 years. For the Dutch Rhine river delta (used as a case study in this paper), water levels and related discharges have been registered since 1901 while design discharges have a return period up to 100,000 years (Van der Most et al., 2014). Extrapolation of these measured discharges to such return periods results in large confidence intervals of the predicted design discharges. Uncertainty in the design discharges used for flood risk assessment can have major implications for national flood protection programs since it determines whether and where dike reinforcements are required. A too wide uncertainty range may lead to unnecessary investments.

To obtain an estimation of a flood with a return period of e.g. 10,000 years with little uncertainty, a discharge data set of at least 100,000 years is required (Klemeš, 1986). Of course, such data sets do not exist. For this reason, many studies try to extend

the data set of measured discharges with historic and/or paleo flood events. The most common methods in literature to include historical data into an FFA are based on the traditional methods of frequentist statistics (Frances et al., 1994; MacDonald et al., 2014; Sartor et al., 2010) and Bayesian statistics (O'Connell et al., 2002; Parkes and Demeritt, 2016; Reis and Stedinger, 2005).

While frequentist statistics are generally applied by decision makers, Bayesian statistics have significantly increased in popularity in the last decade. Reis and Stedinger (2005) has successfully applied a Bayesian Markov Chain Monte Carlo (MCMC) analysis to determine flood frequency relations and their uncertainties using both systematic data and historic flood events. A Bayesian analysis determines the full posterior distribution of the parameters of a probability distribution function (e.g. GEV distribution). This has as advantage that the entire range of parameter uncertainty can be included in the analysis. Contrarily, classical methods based on frequentist statistics usually only provide a point estimate of the parameters where after their uncertainties are commonly described by using the assumption of symmetric normal distributed uncertainty intervals (Reis and Stedinger, 2005). The study of Reis and Stedinger (2005) shows that confidence intervals of design discharges were reduced significantly by extending the systematic data set with historic events using the proposed Bayesian framework. This finding is important for the design of future flood reducing measures since these can then be designed with less uncertainty.

However, Bayesian statistic also has several drawbacks. Although no assumption about the parameter uncertainty of the distribution function has to be made, the results depend on the parameter priors which have to be chosen a priori. The influence of the priors on the posterior distributions of the parameters and hence on the uncertainty of flood frequency relations can even be larger than the influence of discharge measurement errors (Neppel et al., 2010). The prior can be estimated by fitting the original data with the use of e.g. the Maximum Likelihood method. However, we do not have any measurements in, or near to, the tail of the frequency distribution functions. In this way, the benefits of the Bayesian method compared to a traditional flood frequency analysis are at least questionable.

In this study, we propose a systematic approach to include historic flood information into flood safety assessments. The general methodology of a flood frequency analysis remains, only the data set of measured discharges is extended with the use of a bootstrap approach. As a result, this method is close to current practice of water managers. We extend the data set of measured discharges at Lobith, the German-Dutch border, with historic events to decrease uncertainty intervals of design discharges corresponding to rare events. A bootstrap method is proposed to create a continuous data set after which we perform a traditional FFA to stay in line with the current methods used for Dutch water policy. Hence, the results are well understandable by decision makers since solely the effect of using data sets with different lengths on flood frequency relations and corresponding uncertainty intervals are presented. The objective of this study is thus to develop a straightforward method to consider historic flood events in an FFA, while the basic principles of an FFA remain unchanged.

The measured discharges at Lobith (1901-2018) are extended with the continuous reconstructed data set of Toonen (2015) covering the period 1772-1900. These data sets are extended with the most extreme, older historic flood events near Cologne reconstructed by Meurs (2006), which are routed towards Lobith. For this routing, a one dimensional-two dimensional (1D-2D) coupled hydraulic model is used to determine the maximum discharges during these historic events based on the current geometry. In such a way, the historic floods are corrected for anthropogenic interventions and natural changes of the river system, referred to as *normalization* in this study. Normalizing the historic events is of high importance since flood patterns

most likely change over the years as a result of e.g. dike reinforcements, land use change or decrease in floodplain area (dike shifts). The normalized events almost always lead to a higher discharge than the historic event. This is because more water is capable of flowing through the river system as a result of the heightened dikes along the Lower Rhine. Nowadays, floods occur for higher discharge stages compared to the historical time period. In any case, the normalized events give insight in the consequences of an event with the same characteristics of a historic flood event translated to present times. To create a continuous data set, a bootstrap resampling technique is used. The results of the bootstrap method are evaluated against an FFA based on solely measured annual maximum discharges (1901-2018 and 1772-2018). Specifically, the change in the design discharge and its 95% confidence interval of events with a return period of 100,000 years is considered because this design discharge corresponds with the highest safety level used in Dutch flood protection programs (Van Alphen, 2016).

In Section 2 the different data sets used to construct the continuous discharge data set are explained, as well as the 1D-2D coupled hydraulic model. Next, the bootstrap method and FFA are explained (Section 3 and Section 4 respectively). After that, the results of the FFA are given (Section 5). The paper ends with a discussion (Section 6) and the main conclusions (Section 7).

## 2 Annual maximum discharges

### 2.1 Discharge measurements period 1901 - present

Daily discharge observations at Lobith have been performed since 1901 and are available at https://waterinfo.rws.nl. From this data set, the annual maximum discharges are selected in which the hydrologic time period, starting at the $1^{st}$ of October and ending at the $30^{th}$ of September, is used. Since changes to the system have been made the last century, Tijssen (2009) has normalized the measured data set from 1901-2008 for the year 2004. In the $20^{th}$ century, canalization projects were executed along the Upper Rhine (Germany) which were finalized in 1977 (Van Hal, 2003). After that, retention measures were executed in the trajectory Andernach-Lobith. Firstly, the 1901-1977 data set has been normalized with the use of a regression function describing the influence of the canalization projects on the maximum discharges. Then, again a regression function was used to normalize the 1901-2008 data set for the retention measures (Van Hal, 2003). This results in a normalized 1901-2008 data set for the year 2004. For the period 2009-2018, the measured discharges without normalization are used.

During the discharge recording period, different methods have been used to perform the measurements. These different methods result in different uncertainties (Table 1 and must be included in the FFA to correctly predict the 95% confidence interval of the FF curve. From 1901 until 1950, discharges at Lobith were based on velocity measurements performed with floating sticks on the water surface. Since the velocity was only measured at the surface, extrapolation techniques were used to compute the total discharge. This resulted in an uncertainty of approximately 10% (Toonen, 2015). From 1950 until 2000, current meters were used to construct velocity-depth profiles. These profiles were used to compute the total discharge, having an uncertainty of approximately 5% (Toonen, 2015). Since 2000, Acoustic Doppler Current Profiles have been used for which also an uncertainty of 5% is assumed.

**Table 1.** Uncertainties and properties of the various data sets used. The 1342-1772 data set represents the historical discharges (first row in the table), whereas the data sets in the period 1772-2018 are referred to as the systematic data set (rows 2-7)

| Time period | Data source | Property | Cause uncertainty | Location |
|---|---|---|---|---|
| 1342-1772 | Meurs (2006) | 12 single events | Reconstruction uncertain caused by main channel bathymetry, bed friction and maximum occurred water levels | Cologne |
| 1772-1865 | Toonen (2015) | Continuous data set | Reconstruction uncertainty based on measured water levels of surrounding sites ($\sim 12\%$) | Emmerich, Pannerden and Nijmegen |
| 1866-1900 | Toonen (2015) | Continuous data set | Uncertainty caused by translation measured water levels into discharges ($\sim 12\%$) | Lobith |
| 1901-1950 | Tijssen (2009) | Continuous data set | Uncertainty caused by extrapolation techniques to translate measured velocities at the water surface into discharges (10%) | Lobith |
| 1951-2000 | Tijssen (2009) | Continuous data set | Uncertainty caused by translation velocity-depth profiles into discharges (5%) | Lobith |
| 2001-2008 | Tijssen (2009) | Continuous data set | Measurement errors (5%) | Lobith |
| 2009-2018 | Measured water levels available at https://waterinfo.rws.nl | Continuous data set | Measurement errors (5%) | Lobith |

## 2.2 Water level measurements period 1772 - 1900

Toonen (2015) studied the effects of non-stationarity in flooding regimes over time on the outcome of an FFA. He extended the data set of measured discharges of the Rhine river at Lobith with the use of water level measurements. At Lobith, daily water level measurements are available since 1866. For the period 1772-1865 water levels were measured at the nearby gauging locations Emmerich, Germany (located 10 kilometers in upstream direction), Pannerden (located 10 kilometers in downstream direction) and Nijmegen (located 22 kilometers in downstream direction). Toonen (2015) used the water levels of these locations to compute the water levels at Lobith and their associated uncertainty interval with the use of a linear regression between the different measurement locations. Subsequently, he translated these water levels, together with the measured water levels for the period 1866-1900, into discharges using stage-discharge relations at Lobith. These relations were derived based on discharge predictions adopted from Cologne before 1900 and measured discharges at Lobith after 1900, and water level estimates from the measurement locations Emmerich, Pannerden, Nijmegen and Lobith. Since the discharge at Cologne strongly correlates with the discharge at Lobith, the measured discharges in the period 1817-1900 could be used to predict discharges

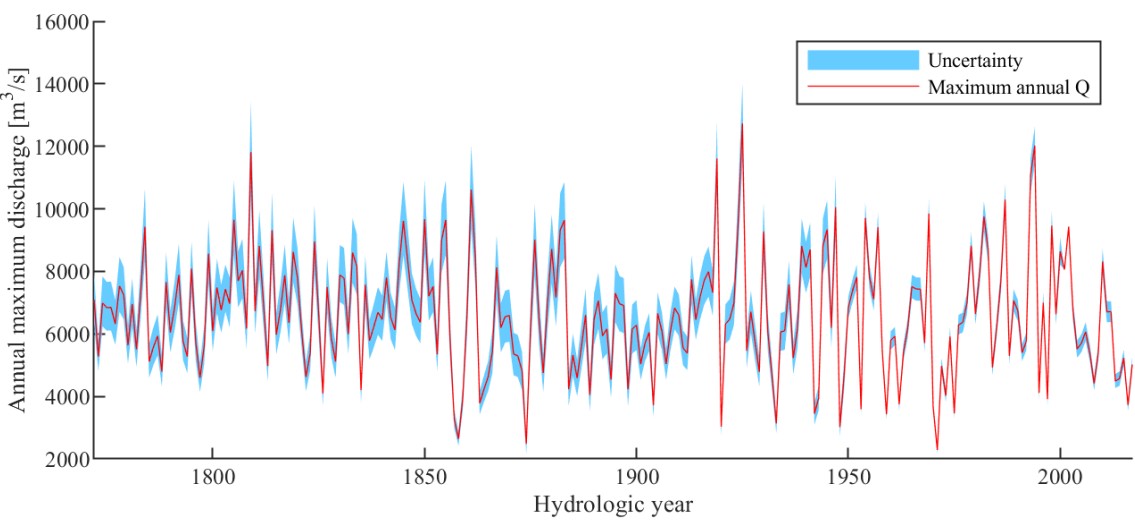

**Figure 1.** Maximum discharges (Q) and their 95% confidence interval during the systematic time period (1772-2018)

at Lobith. The 95% confidence interval in reconstructed water levels propagates in the application of stage-discharge relations, resulting in an uncertainty range of approximately 12% for the reconstructed discharges (Fig. 1) (Toonen, 2015).

The reconstructed discharges in the period 1772-1900 represent the computed maximum discharges at the time of occurrence and these have not been normalized for changes in the river system. They thus represent the actual occurred annual maximum discharges. Toonen (2015) argues that, based on the work of Bronstert et al. (2007) and Vorogushyn and Merz (2013), the effect of recent changes in the river system on discharges of extreme floods of the Lower Rhine is small. Hence, it is justified to use the presented data set of Toonen (2015) in this study as normalized data. Fig. 1 shows the annual maximum discharges for the period 1772-2018 and their 95% confidence intervals. This data represents the systematic data set and consists of the measured discharges covering the period 1901-2018 and the reconstructed data set of Toonen (2015) covering the period 1772-1900.

## 2.3 Reconstructed flood events period 1300 AD - 1772

Meurs (2006) has reconstructed maximum discharges during historic flood events near the city of Cologne, Germany. The oldest event dates back to 1342. Only flood events caused by high rainfall intensities were reconstructed because of the different hydraulic conditions of flood events caused by ice jams. The used method is described in detail by Herget and Meurs (2010), in which the 1374 flood event was used as a case study. Historic documents providing information about the maximum water levels during the flood event were combined with the reconstruction of the river cross section at that same time. Herget and Meurs (2010) calculated mean flow velocities near the city of Cologne at the time of the historic flood events with the use of the Manning's equation:

$$Q_\mathrm{p} = A_\mathrm{p} R_\mathrm{p}^{2/3} S^{1/2} n^{-1} \qquad (1)$$

where $Q_p$ represents the peak discharge (m³/s), $A_p$ the cross-sectional area (m²) during the highest flood level, $R_p$ the hydraulic radius during the highest flood level (m), $S$ the slope of the main channel and $n$ its Manning's roughness coefficient (s/m$^{1/3}$). However, the highest flood level as well as the Manning's roughness coefficient are uncertain. The range of maximum water levels were based on historical sources, whereas the range of Manning's roughness coefficients were based on the tables of Chow (1959). Including these uncertainties in the analysis, Herget and Meurs (2010) were able to calculate maximum discharges of the specific historic flood events and associated uncertainty ranges (Fig. 4).

In total 13 historic flood events that occurred before 1772 were reconstructed. Two of the flood events occurred in 1651. Only the largest flood of these two is considered as data point. This results in 12 historic floods that are used to extend the systematic data set. The reconstructed maximum discharges at Cologne (Meurs, 2006) are used to predict maximum discharges at Lobith with the use of a hydraulic model to normalize the data set. Although Cologne is located roughly 160 km upstream of Lobith, there is a strong correlation between the discharges at these two locations. This is because they are located in the same fluvial trunk valley and only have minor tributaries (Sieg, Ruhr and Lippe rivers) joining in between (Toonen, 2015). This makes the reconstructed discharges at Cologne applicable to predict corresponding discharges at Lobith. The model used to perform the hydraulic calculations is described in Section 2.3.1. The maximum discharges at Lobith of the 12 historic flood events are given in Section 2.3.2.

### 2.3.1 Model environment

In this study, the 1D-2D coupled modelling approach as described by Bomers et al. (2019a) is used to normalize the data set of Meurs (2006). This normalization is performed by routing the reconstructed historical discharges at Cologne over modern topography to estimate the maximum discharge at Lobith in present times. The study area stretches from Andernach to the Dutch cities of Zutphen, Rhenen and Druten (Fig. 2). In the hydraulic model, the main channels and floodplains are discretized by 1D profiles. The hinterland is discretized by 2D grid cells. The 1D profiles and 2D grid cells are connected by a structure corresponding with the dimensions of the dike that protects the hinterland from flooding. If the computed water level of a 1D profile exceeds the dike crest, water starts to flow into the 2D grid cells corresponding with inundations of the hinterland. A discharge wave is used as upstream boundary condition. Normal depths, computed with the use of the Manning's equation, were used as downstream boundary conditions. HEC-RAS (v. 5.0.3) (Brunner, 2016), developed by the Hydrologic Engineering Centre (HEC) of the US Army Corps of Engineers, is used to perform the computations. For more information about the model set-up, see Bomers et al. (2019b).

### 2.3.2 Normalization historic flood events

We use the hydraulic model to route the historical discharges at Cologne, as reconstructed by Meurs (2006), to Lobith. However, the reconstructed historical discharges were uncertain. Therefore, also the discharges at Lobith are uncertain. To include this uncertainty in the analysis a Monte Carlo analysis (MCA) is performed in which, among others, the upstream discharges reconstructed by Meurs (2006) are included as random parameters. These discharges have large confidence intervals (Fig. 4). The severe 1374 flood, representing the largest flood of the last 1,000 years with a discharge of 23,000 m³/s, even has a

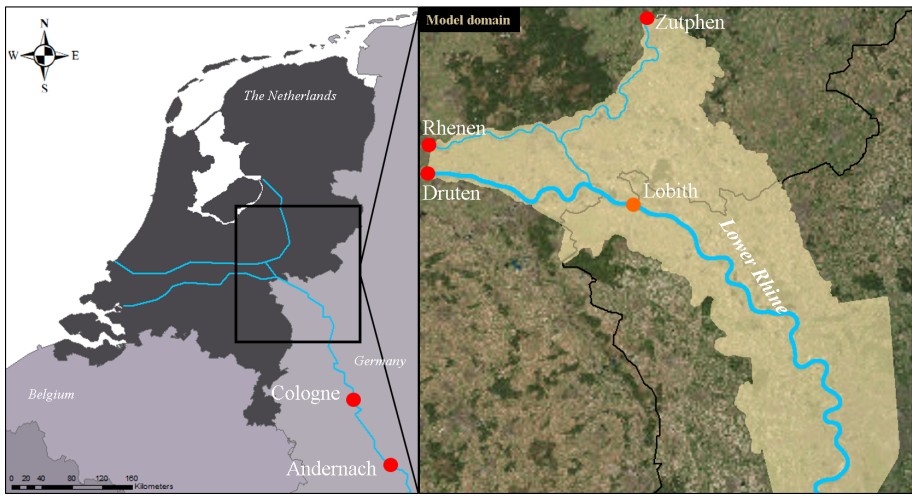

**Figure 2.** Model domain (blue river branches) of the 1D-2D coupled model

confidence interval of more than 10,000 m³/s. To include the uncertainty as computed by Meurs (2006) in the analysis, the maximum upstream discharge is varied in the MCA based on its probability distribution. However, the shape of this probability distribution is unknown. Herget and Meurs (2010) only provided the maximum, minimum and mean value of the reconstructed discharges. We assumed normally distributed discharges since it is likely that the mean value has a higher probability of

occurrence than the boundaries of the reconstructed discharge range. However, we found that the assumption of the uncertainty distribution has a negligible effect on the 95% uncertainty interval of the FF curve at Lobith. Assuming uniformly distributed uncertainties only led to a very small increase in this 95% uncertainty interval.

Not only the maximum discharges at Cologne are uncertain, also the discharge wave shape of the flood event. The shape of the upstream flood event may influence the maximum discharge at Lobith. Therefore, the upstream discharge wave shape

is varied in the MCA. We use a data set of approximately 250 potential discharge wave shapes that can occur under current climate conditions (Hegnauer et al., 2014). In such a way, a broad range of potential discharge wave shapes, e.g. a broad peak, a small peak, or two peaks, are included in the analysis. For each run in the MCA, a discharge wave shape is randomly sampled and scaled to the maximum value of the flood event considered (Fig. 3). This discharge wave represents the upstream boundary condition of the model run.

The sampled upstream discharges, based on the reconstructed historic discharges at Cologne, may lead to dike breaches in present times. Since we are interested in the consequences of the historic flood events in present times, we want to include these dike breaches in the analysis. However, it is highly uncertain how dike breaches develop. Therefore, the following potential dike breach settings are included in the MCA (Fig. 3):

1. Dike breach threshold

2. Final dike breach width

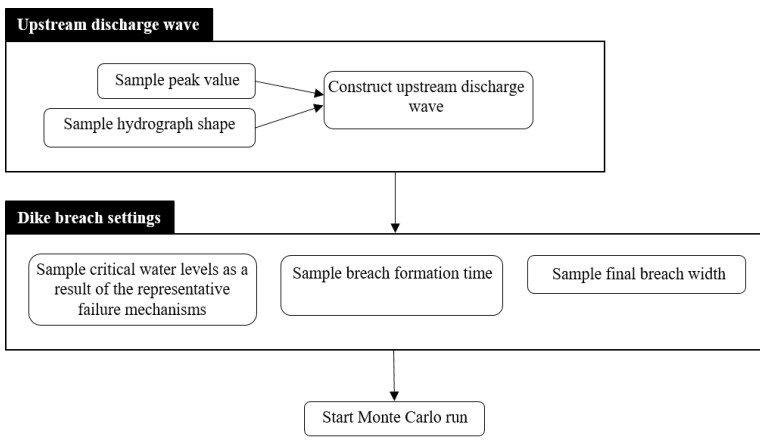

**Figure 3.** Random input parameters considered in the Monte Carlo analysis

3. Dike breach duration

The dike breach thresholds (i.e. the critical water level at which a dike starts to breach) are based on 1D fragility curves provided by the Dutch Ministry of Infrastructure and Water Management. A 1D fragility curve expresses the reliability of a flood defence as a function of the critical water level (Hall et al., 2003). The critical water levels thus influence the timing of dike breaching. For the Dutch dikes, it is assumed that the dikes can fail due to failure mechanisms wave overtopping and overflow, piping and macro-stability, where as the German dikes only fail because of wave overtopping and overflow (Bomers et al., 2019b). The distributions of the final breach width and the breach formation time are based on literature and on historical data (Apel et al., 2008; Verheij and Van der Knaap, 2003). Since it is unfeasible to implement each dike kilometer as potential dike breach location in the model, only the dike breach locations that result in significant overland flow are implemented. This results in 33 potential dike breach locations whereas overflow (without dike breaching) is possible to occur at every location throughout the model domain (Bomers et al., 2019b).

So, for each Monte Carlo run an upstream maximum discharge and discharge wave shape is sampled. Next, for each of the 33 potential dike breach locations the critical water level, dike breach duration and final breach widths are sampled. With this data, the Monte Carlo run representing a specific flood scenario can be run (Fig. 3). This process is repeated until converge of the maximum discharge at Lobith and its confidence interval is found. For a more in depth explanation of the Monte Carlo analysis and random input parameters, we refer to Bomers et al. (2019b).

The result of the MCA is the normalized maximum discharge at Lobith and its 95% confidence interval for each of the 12 historic flood events. Since the maximum discharges at Cologne are uncertain, also the normalized maximum discharges at Lobith are uncertain (Fig. 4). Fig 4 shows that the extreme 1374 flood with a maximum discharge of between 18,800 m³/s and 29,000 m³/s at Cologne, reduces significantly in downstream direction as a result of overflow and dike breaches. Consequently, the maximum discharge at Lobith turns out to be between 13,825 and 17,753 m³/s. This large reduction in the maximum discharge is caused by the major overflow and dike breaches that occur in present times. Since the 1374 flood event was much

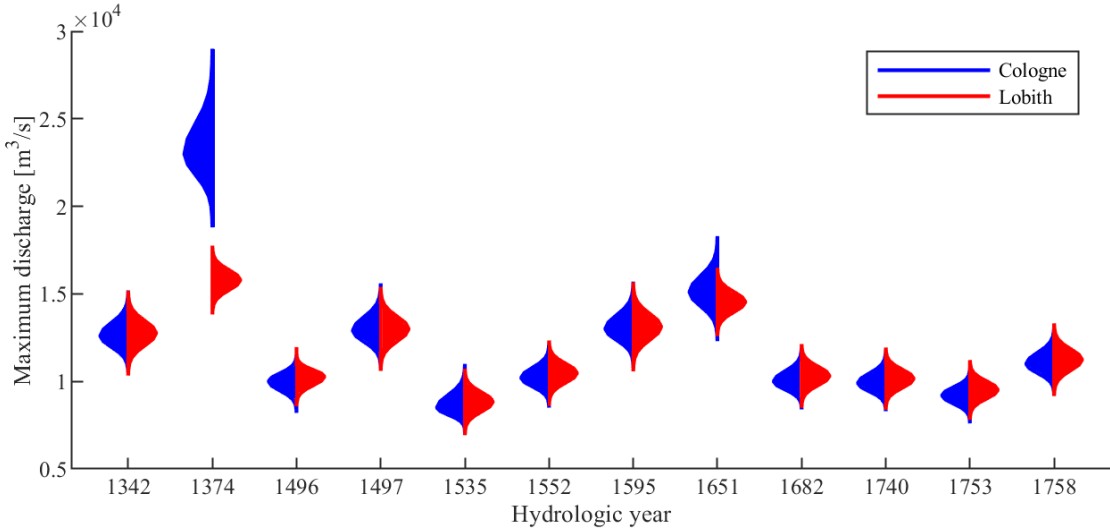

**Figure 4.** Maximum discharges and their 95% confidence intervals of the reconstructed historic floods at Cologne (Herget and Meurs, 2010) and simulated maximum discharges and their 95% confidence intervals at Lobith for the 12 historic flood events

larger than the current discharge capacity of the Lower Rhine, the maximum discharge at Lobith decreases. The reconstruction of the 1374 flood over modern topography is presented in detail in Bomers et al. (2019c). On the other hand, the other 11 flood events were below this discharge and hence only a slight reduction in discharges was found for some of the events as a result of dike breaches whereas overflow did not occur. Some other events slightly increased as a result of the inflow of the tributaries

5   Sieg, Ruhr and Lippe rivers along the Lower Rhine. This explains why the 1374 flood event is much lower at Lobith compared to the discharge at Andernach, while the discharges of the other 11 flood events are more or less the same at these two locations (Fig. 4). The reduction in maximum discharge of the 1374 flood event in downstream direction shows the necessity to apply hydraulic modelling since the use of a linear regression analysis based on measured discharges between Cologne and Lobith will result in an unrealistic larger maximum discharge at Lobith.

10   The reconstructed discharges at Lobith are used to extend the systematic data set presented in Fig. 1. In the next section, these discharges are used in an FFA with the use of a bootstrap method.

## 3   Bootstrap method

The systematic data set covering the period 1772-2019 is extended with 12 reconstructed historic flood events that occurred in the period 1300-1772. To create a continuous data set, a bootstrap method based on sampling with replacement is used. The

15   continuous systematic data set (1772-2018) is resampled over the missing years from the start of the historical period to the start of the systematic record. Two assumptions must be made such that the bootstrap method can be applied:

1. The start of the continuous discharge series since the true length of the historical period is not known.

2. The perception threshold over which floods were recorded in the historical times before water level and discharge measurements were conducted.

Assuming that the historical period starts with the first known flood (in this study: 1342) will significantly underestimate the true length of this period. This underestimation influences the shape of the FF curve (Hirsch and Stedinger, 1987; Schendel and Thongwichian, 2017). Therefore, Schendel and Thongwichian (2017) proposed the following equation to determine the length of the historical period:

$$M = L + \frac{L + N - 1}{k} \tag{2}$$

where $M$ represents the length of the historical period (years), $L$ the number of years from the first historic flood to the start of the systematic record (431 years), $N$ the length of the systematic record (247 years) and $k$ the number of floods exceeding the perception threshold in both the historical period as well as in the systematic record (28 in total). Using equation 2 results in a length of the historical period of 455 years (1317-1771).

The perception threshold is considered to be equal to the discharge of the smallest flood present in the historic period, representing the 1535 flood with an expected discharge of 8,826 m$^3$/s (Fig. 4). We follow the method of Parkes and Demeritt (2016) assuming that the perception threshold was fairly constant over the historical period. However, the maximum discharge of the 1535 flood is uncertain and hence also the perception threshold is uncertain. Therefore, the perception threshold is treated as a random uniformly distributed parameter in the bootstrap method which boundaries are based on the 95% confidence interval of the 1535 flood event.

The bootstrap method consist of creating a continuous discharge series from 1317-2018. The method includes the following steps (Fig. 5):

1. Combine the 1772-1900 data set with the 1901-2018 data set to create a systematic data set.

2. Select the flood event with the lowest maximum discharge present in the historic time period. Randomly sample a value in between the 95% confidence interval of this lowest flood event. This value is used as perception threshold.

3. Compute the start of the historical time period (equation 2).

4. Of the systematic data set, select all discharges that have an expected value lower than the sampled perception threshold.

5. Use the data set created in Step 4 to create a continuous discharge series in the historical time period. Randomly draw an annual maximum discharge of this systematic data set for each year within the historical period of which no data is available following a bootstrap approach.

6. Since both the reconstructed as well as the measured discharges are uncertain due to e.g. measurement errors, these uncertainties must be included in the analyses. Therefore, for each discharge present in the systematic data set and in the historical data set, its value is randomly sampled based on its 95% confidence interval.

7. Combine the data sets of Steps 5 and 6 to create a continuous data set starting from 1317-2018.

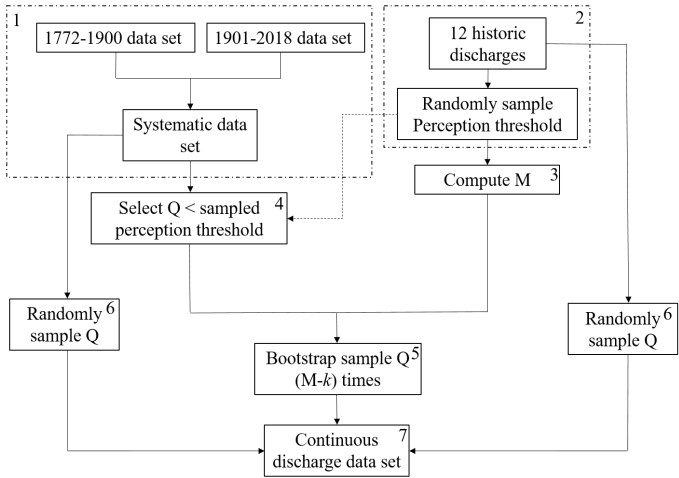

**Figure 5.** Bootstrap method to create a continuous discharge series

The presented steps in the bootstrap method are repeated 5,000 times in order to create 5,000 continuous discharge data sets resulting in convergence in the FFA. The FFA procedure itself is explained in the next section.

## 4   Flood frequency analysis

An FFA is performed to determine the FF relation of the different data sets (e.g. systematic record, historical records). A probability distribution function is used to fit the annual maximum discharges to its probability of occurrence. Many types of distribution functions and goodness-of-fit tests exist, all with their own properties and drawbacks. However, the available goodness-of-fit tests for selecting an appropriate distribution function are often inconclusive. This is mainly because each test is more appropriate for a specific part of the distribution, while we are interested in the overall fit since the safety standards expressed in probability of flooding along the Dutch dikes vary from $10^{-2}$ to $10^{-5}$. Furthermore, we highlight that we focus on the influence of extending the data set of measured discharges on the reduction in uncertainty of the FF relations rather than on the suitability of the different distributions and fitting methods.

We restrict our analysis to the use of a Generalized Extreme Value (GEV) distribution since this distribution is commonly used in literature to perform an FFA (Parkes and Demeritt, 2016; Haberlandt and Radtke, 2014; Gaume et al., 2010). Additionally, several studies have shown the applicability of this distribution on the flooding regime of the Rhine river (Toonen, 2015; Chbab et al., 2006; Te Linde et al., 2010). The GEV distribution has an upper bound and is thus capable of flattening off at extreme values by having a flexible tail. We use a bounded distribution since the maximum discharge that is capable of entering the Netherlands is limited to a physical maximum value. The crest levels of the dikes along the Lower Rhine, Germany, are not infinitely high. The height of the dikes influences the discharge capacity of the Lower Rhine and hence the discharge that can

**Table 2.** Discharges [m³/s] and their 95% confidence interval corresponding to several return periods for the 1901, 1772 and 1317 data sets and the data set of Toonen (2015)

| Data | $Q_{10}$ | $Q_{100}$ | $Q_{1,000}$ | 2.5% | $Q_{1,250}$ | 97.5% | 2.5% | $Q_{100,000}$ | 97.5% |
|------|------|------|------|------|------|------|------|------|------|
| 1901-2018 | 9,264 | 12,036 | 14,050 | 10,594 | 14,215 | 20,685 | 11,301 | 16,649 | 29,270 |
| 1772-2018 | 9,106 | 11,442 | 13,008 | 11,053 | 13,130 | 16,027 | 11,858 | 14,813 | 19,576 |
| 1317-2018 | 8,899 | 11,585 | 13,655 | 12,514 | 13830 | 15,391 | 14,424 | 16,562 | 19,303 |

flow towards Lobith. Using an upper bounded distribution yields that the FF relation converges towards a maximum value for extremely large return periods. This value represents the maximum discharge that is capable of occurring at Lobith.

The GEV distribution is described with the following equation:

$$F(x) = exp\{-[\xi\frac{x-\mu}{\sigma}]^{\frac{1}{\xi}}\}$$ (3)

where $\mu$ represents the location parameter indicating where the origin of the distribution is positioned, $\sigma$ the scaling parameter describing the spread of the data, and $\xi$ represents the shape parameter controlling the skewness and kurtosis of the distribution, both influencing the upper tail and hence the upper bound of the system. The maximum likelihood method is used to determine the values of the three parameters of the GEV distribution (Stendinger and Cohn, 1987; Reis and Stedinger, 2005).

The FFA is performed for each of the 5,000 continuous discharge data sets created with the bootstrap method (Section 3), resulting in 5,000 fitted GEV curves. The average of these relations is taken to get the final FF curve and its 95% confidence interval. The results are given in the next section.

## 5 Results

### 5.1 Flood frequency relations

In this section the FFA results (Fig. 6) of the following data sets are presented:

- 1901 data set; measured discharges covering the period 1901-2018.

- 1772 data set; as above and extended with the data set of Toonen (2015), representing the systematic data set and covering the period 1772-2018.

- 1317 data set; as above and extended with 12 reconstructed historic discharges and the bootstrap resampling method to create a continuous discharge series covering the period 1317-2018.

If the data set of measured discharges is extended, we find a large reduction in the confidence interval of the FF curve (Fig. 6 and Table 2). Only extending the data set with the data of Toonen (2015) reduced this confidence interval with 5,200 m³/s for the floods with a return period of 1,250 years (Table 2). Adding the reconstructed historic flood events in combination with

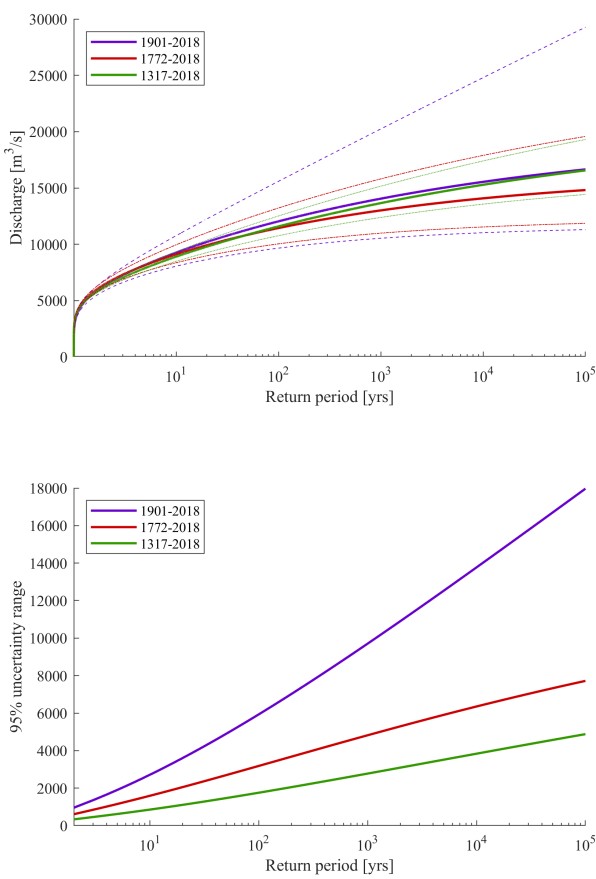

**Figure 6.** Fitted GEV curves and their 95% confidence intervals of the 1901, 1772 and 1317 data sets

a bootstrap method to create a continuous data set, results in an even larger reduction in the confidence interval of 7,400 m³/s compared to the results of the 1901 data set. For the discharges with a return period of 100,000 years, we find an even larger reduction in the confidence intervals (Table 2).

    Furthermore, we find that using only the 1901 data set results in larger design discharges compared to the two extended
5  data sets. This is in line with the work of Toonen (2015). Surprisingly however, we find that the 1772 data set predicts the lowest discharges for return periods > 100 years (Table 2), while we would expect that the 1317 data set predicts the lowest values according to the findings of Toonen (2015). The relatively low positioning of the FF curve constructed with the 1772 data, compared to our other 1317 and 1901 data sets, might be explained by the fact that the data of Toonen (2015) covering the period 1772-1900 has not been normalized. This period has a relative high flood intensity (Fig. 1). However, only two
10  flood events exceeded 10,000 m³/s. A lot of dike reinforcements along the Lower Rhine were executed during the last century. Therefore, it is likely that before the 20[th] century, flood events with a maximum discharge exceeding 10,000 m³/s resulted in

dike breaches and overflow upstream of Lobith. As a result, the maximum discharge of such an event decreased significantly. Although Toonen (2015) mentions that the effect of recent changes in the river system on discharges of extreme floods of the Lower Rhine is small, we argue that it does influence the flood events with maximum discharges slightly lower than the current main channel and floodplains capacity. Currently, larger floods are possible to flow in downstream direction without

the occurrence of inundations compared to the 19th century. Therefore, it is most likely that the 1772-1900 data set of Toonen (2015) underestimates the flooding regime of that specific time period influencing the shape of the FF curve.

## 5.2  Hypothetical future extreme flood event

After the 1993 and 1995 flood events of the Rhine river, the FF relation used in Dutch water policy was recalculated taking into account the discharges of these events. All return periods were adjusted. The design discharges with a return period of

1,250 years, which was the most important return period at that time, increased with 1,000 m$^3$/s (Parmet et al., 2001). Such an increase in the design discharge requires more investments in dike infrastructure and floodplain measures to re-establish the safety levels. Parkes and Demeritt (2016) found similar results for the river Eden, UK. They showed that the inclusion of the 2015 flood event had a significant effect on the upper tail of the FF curve, even though their data set was extended from 1967 to 1800 by adding 21 reconstructed historic events to the data set of measured data. Schendel and Thongwichian (2017)

argues that if the flood frequency relation alters after a recent flood, and if this change can be ambiguously attributed to this event, the data set of measured discharges must expanded since otherwise the FF results will be upward biased. Based on their considerations, it is interesting to see how adding a single extreme flood event influences the results of our method.

Both the 1317 and 1901 data sets are extended from 2018 to 2019 with a hypothesized flood in 2019. We assume that in 2019 a flood event has occurred that equals the largest measured discharge so far. This corresponds with the 1926 flood event

(Fig. 1), having a maximum discharge of 12,600 m$^3$/s. No uncertainty of this event is included in the analysis. Fig. 7 shows that the FF curve based on the 1901 data set changes significantly as a result of this hypothesized 2019 flood. We calculate an increase in the discharge corresponding with a return period of 100,000 years of 1,280 m$^3$/s. Contrarily, the 2019 flood has almost no effect on the extended 1317 data set. The discharge corresponding to a return period of 100,000 years only increased slightly with 180 m$^3$/s. Therefore, we conclude that the extended data set is more robust to changes in FF relations as a result

of future flood events. Hence, we expect that the changes in FF relations after the occurrence of the 1993 and 1995 flood events would be less severe if the analysis was performed with an extended data set as presented in this study. Consequently, decision makers might have taken a different decision since less investments were required to cope with the new flood safety standards. Therefore, we recommend to use historical information about the occurrence of flood events in future flood safety assessments.

## 6  Discussion

We developed an efficient bootstrap method to include historic flood events in an FFA. We used a 1D-2D coupled hydraulic model to normalize the data set of Meurs (2006) for modern topography. An advantage of the proposed method is that any kind of historical information (e.g. flood marks, sediment depositions) can be used to extend the data set of annual maximum

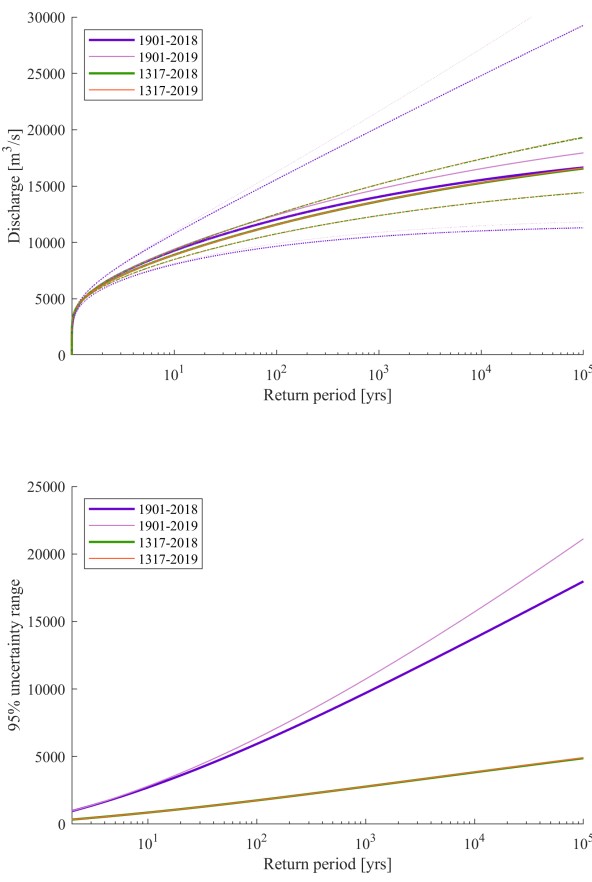

**Figure 7.** Fitted GEV curves and their 95% confidence intervals of the 1901 and 1317 data sets if they are extended with a future flood event

discharges as long as the information can be translated into discharges. Another great advantage of the proposed method is the computational time to create the continuous data sets and to fit the GEV distributions. The entire process is completed within several minutes. Furthermore, it is easy to update the analysis if more historical information about flood events becomes available. However, the method is based on various assumptions and has some drawbacks. These assumptions and drawbacks are discussed below.

## 6.1 Added value of normalized historic flood events

The results have shown that extending the systematic data set with normalized historic flood events can significantly reduce the confidence intervals of the FF curves. This is in line with the work of O'Connell et al. (2002) who claim that the length of the instrumental record is the single most important factor influencing uncertainties in flood frequency relations. However, reconstructing historic floods is time consuming, especially if these floods are normalized with a hydraulic model. Therefore,

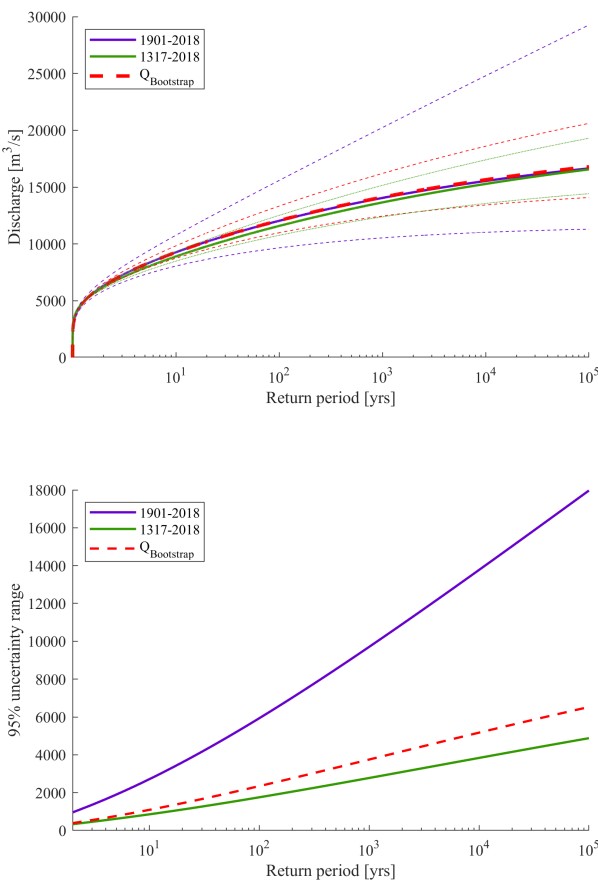

**Figure 8.** Fitted GEV curves of the 1901, 1317 and $Q_{Bootstrap}$ data sets

the question arises whether it is required to reconstruct historic floods to extend the data set of measured discharges. Another, less time consuming, option might be to solely resample the measured discharges in order to extend the length of the data set. Such a method was applied by Chbab et al. (2006) who resampled 50 years of weather data to create a data set of 50,000 years of annual maximum discharges.

5     To test the applicability of solely using measured discharges, we use the bootstrap method presented in Section 3. A data set of approximately 700 years (equal to the length of the 1317 data set) is created based on solely measured discharges in the period 1901-2018. The perception threshold is assumed to be equal to the lowest measured discharge such that the entire data set of measured discharges is used during the bootstrap resampling. Again, 5,000 discharge data sets are created to reach convergence in the FFA. This data is referred to as the $Q_{Bootstrap}$ data set.

10     We find that the use of the $Q_{Bootstrap}$ data set, based on solely resampling the measured discharges of the 1901 data set, results in lower uncertainties of the FF curve compared to the 1901 data set (Fig. 8). This is because the length of the measured data

set is increased through the resampling method. Although the confidence interval decreases after resampling, the confidence interval of the $Q_{Bootstrap}$ data set is still larger compared to the 1317 data set, including the normalized historic flood events (Fig. 8). This is because the variance of the $Q_{Bootstrap}$ data set, which is equal to 4.19 x $10^6$ m$^3$/s, is still larger than the variance of the 1317 data set. For the $Q_{Bootstrap}$ data set, the entire measured data set (1901-2018) is used for resampling, while for the 1317 data set only the discharges below a certain threshold in the systematic time period (1772-2018) are used for resampling. The perception threshold was chosen to be equal to the lowest flood event in the historical time period having a discharge of between 6,928-10,724 m$^3$/s. Hence, the missing years in the historical time period are filled with relatively low discharges. Therefore, the variance of the 1317 data set is relatively low (3.35 x $10^6$ m$^3$/s) as a result of the lower discharges to create the continuous data set. As a result of the lower variance, also the uncertainty intervals are smaller compared to the $Q_{Bootstrap}$ data set.

Furthermore, the FF curve of the $Q_{Bootstrap}$ data set is only based on a relatively short data set of measured discharges and hence only based on the climate conditions of this period. Extending the data set with historic flood events gives a better representation of the long-term climatic variability in flood events since these events only have been normalized for changes in the river system and thus still capture the climate signal. We conclude that reconstructing historic events, even if their uncertainty is large, is worth the effort since it reduces the uncertainty intervals of design discharges corresponding to rare flood events which is crucial for flood protection policy-making.

## 6.2 Resampling systematic data set

The shape of the constructed FF curve strongly depends on the climate conditions of the period considered. If the data set is extended with a period which only has a small number of large flood events, this will result in a significant shift of the FF curve in downward direction. This shift can be overestimated if the absence of large flood events only applies to the period used to extend the data set. Furthermore, by resampling the measured data set, we assume that the flood series consist of independent and identically distributed random variables. This might not be the case if climate variability plays a significant roll in the considered time period resulting in a period of e.g. extreme low or high flows. However, up till now no consistent large-scale climate change signal in observed flood magnitudes has been identified (Blöschl et al., 2017).

In Section 5, we found that extending the data set from 1901 to 1772 resulted in a shift in downward direction of the FF curve. This is because in the period 1772-1900, a relatively small number of floods exceeded a discharge larger than 10,000 m$^3$/s. Since no large flood events were present in the period 1772-1900, this data set has a lower variance compared to the 1901 data set. Using both the 1772 and 1901 data sets for resampling purposes influences the uncertainty of the FF curve. To identify this effect, we compared the results if solely the measured discharges (1901-2018) are used for resampling purposes and if the entire systematic data set (1772-2018) period is used. We find that using the entire systematic data set results in a reduction in the 95% confidence intervals compared to the situation in which solely the measured discharges are used caused by the lower variance in the period 1772-1900. However, the reduction is at maximum 12% for the return period of 100,000 years. Although the lower variance in the 1772-1900 data set might be explained by the fact that these discharges are not normalized, the lower variance may also be caused by the natural variability in climate.

## 6.3 Distribution function and goodness-of-fit test

In Section 5, only the results for a GEV distribution were presented. We found that the uncertainty interval of the flood event with a return period of 100,000 years was reduced with 73% by extending the data set of approximately 120 years of annual maximum discharges to a data set with a length of 700 years. Performing the analysis with other distributions yield similar results. A reduction of 60% is found for the Gumbel distribution and a reduction of 76% for the Weibull distribution. This shows that, although the uncertainty intervals depend on the probability distribution function used, the general conclusion of reduction in uncertainty of the fitted FF curve holds.

However, by only considering a single distribution function in the analysis, model uncertainty is neglected. One approach to manage this uncertainty is to create a composite distribution of several distributions each allocated a weighting based on how well it fits the available data (Apel et al., 2008). Furthermore, the uncertainty related to the use of various goodness-of-fit tests was neglected since only the Maximum Likelihood function was used to fit the sample data to the distribution function. Using a composite distribution and multiple goodness-of-fit tests will result in an increase in the uncertainties of FF curves.

## 6.4 Length of extended data set and considered perception threshold

The measured data set starting at 1901 was extended to 1317. However, the extended data set still has limited length compared to the maximum return period of 100,000 years considered in Dutch water policy. Preferably, we would like to have a data set with at least the same length as the maximum safety level considered such that extrapolation in FFAs is not required anymore. However, the proposed method is a large step to decrease uncertainty.

Furthermore, the systematic data set was used to create a continuous data set using a bootstrap approach. However, preferably we would like to have a historical continuous record since now the low flows are biased on climate conditions of the last 250 years. Using this data set for resampling influences the uncertainty intervals of the FF curves. If the historical climate conditions highly deviated from the current climate conditions, this approach does not produce a reliable result. In addition, the perception threshold influences the variance of the considered data set and hence the uncertainty of the FF curve. Using a smaller threshold results in an increase in the variance of the data set and hence to an increase in the uncertainty intervals. The proposed assumption related to the perception threshold can only be used if there is enough confidence that the smallest known flood event in the historical time is indeed the actual smallest flood event that occurred in the considered time period.

## 6.5 Comparison with Bayesian statistics

The FFA was performed based on frequentist statistics. The Maximum Likelihood function was used to fit the parameters of the GEV distribution function. However, only point estimates are computed. To enable uncertainty predictions of the GEV parameter estimates, the maximum likelihood estimator assumes symmetric confidence intervals. This may result in an incorrect estimation of the uncertainty which is specifically a problem for small sample sizes. For large sample sizes, maximum likelihood estimators become unbiased minimum variance estimators with approximate normal distributions. Contrarily, Bayesian statistics provide the entire posterior distributions of the parameter estimates and thus no assumptions have to be made. How-

ever, a disadvantage of the Bayesian statistics is that the results are influenced by the priors describing the distributions of the parameters (Neppel et al., 2010). For future work, we recommend to study how uncertainty estimates differ between the proposed bootstrap method and a method which relies on Bayesian statistics such as Reis and Stedinger (2005).

Moreover, a disadvantage of the proposed bootstrap approach is that, by resampling the systematic data set to fill the gaps
in the historical time period, the shape of the flood frequency curve is influenced in the domain corresponding to events with small return periods (i.e. up to $\sim$ 100 years corresponding with the length of the 1901 data set). Methods presented by e.g. Reis and Stedinger (2005) and Wang (1990) use historical information solely to improve the estimation of the tail of the FF curves, while the systematic part of the curve stays untouched. Table 2 shows the discharges corresponding with a return period of 100 years for both the 1901 data set and the extended 1317 data set following the bootstrap method described in Section 3. We find
that this discharge decreases from 12,075 m$^3$/s to 11,628 m$^3$/s by extending the systematic data set. This decrease in design discharge with 3.7% indicates that resampling the systematic data set over the historical time period only has a little effect on the shape of the flood frequency curve corresponding with small return periods justifying the use of the bootstrap method.

## 7   Conclusions

Design discharges are commonly determined with the use of flood frequency analyses (FFA) in which measured discharges are
used to fit a probability distribution function. However, discharge measurements have been performed only for the last 50-100 years. This relatively short data set of measured discharges results in large uncertainties in the prediction of design discharges corresponding to rare events. Therefore, this study presents an efficient bootstrap method to include historic flood events in an FFA. The proposed method is efficient in terms of computational time and set-up. Additionally, the basic principles of the traditional FFA remain unchanged.

The proposed bootstrap method was applied to the discharge series at Lobith. The systematic data set covering the period 1772-2018 was extended with 12 historic flood events. The historic flood events reconstructed by Meurs (2006) had a large uncertainty range, especially for the most extreme flood events. The use of a 1D-2D coupled model reduced this uncertainty range of the maximum discharge at Lobith for most flood events as a result of the overflow patterns and dike breaches along the Lower Rhine. The inclusion of these historic flood events in combination with a bootstrap method to create a continuous
data set, resulted in a decrease in the 95% uncertainty interval of 72% for the discharges at Lobith corresponding to a return period of 100,000 years. Adding historical information about rare events with a large uncertainty range in combination with a bootstrap method has thus the potential to significantly decrease the confidence interval of design discharges of extreme events.

Since correct prediction of flood frequency relations with little uncertainty is of high importance for future national flood protection programs, we recommend to use historical information in FFA. Additionally, extending the data set with historic
events makes the flood frequency relation less sensitive to future flood events. Finally, we highlight that the proposed method to include historical discharges into a traditional FFA can be easily implemented in flood safety assessments because of its simple nature in terms of mathematical computations as well as of its computational efforts.

*Acknowledgements.* This research is supported by the Netherlands Organisation for Scientific Research (NWO, project 14506) which is partly funded by the Ministry of Economic Affairs and Climate Policy. Furthermore, the research is supported by the Ministry of Infrastructure and Water Management and Deltares. This research has benefited from cooperation within the network of the Netherlands Centre for River studies NCR (www.ncr-web.org).

5      The authors would like to thank the Dutch Ministry of Infrastructure and Water Management, Prof. Dr. Herget (University of Bonn) and Dr. Toonen (KU Leuven) for providing the data. Furthermore, the authors would like to thank Dr. Toonen (KU Leuven) for his valuable suggestions that improved the manuscript. In addition, the authors would like to thank Dr. Elena Volpi (Roma Tre University) and the two anonymous reviewer for their suggestions during the discussion period, which greatly improved the quality of the paper. Finally, the authors would like to thank Van der Meulen Msc, Dr. Cohen and Prof. Dr. Middelkoop from Utrecht University for their cooperation in the NWO

10   Project Floods of the past—Design for the future.

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
