# Peer review of "Decreasing uncertainty in flood frequency analyses by including historic flood events in an efficient bootstrap approach"

_Natural Hazards and Earth System Sciences, 2019_

## Referee Comment (RC1) · Elena Volpi (Referee) · 17 Apr 2019

**General comment**

The manuscript investigates how the estimation uncertainty affecting flood frequency curves can be reduced by including historical flood events. The Authors analyses the case of the Dutch Rhine river delta, by using systematic information recorded at the site of interest, systematic information interpolated from close river sections and historical events reconstructed by using hydraulic models. The model is used to "normalize" the historical discharges for anthropogenic and natural changes occurred in the river

system. The study case is o interest for the large amount of hydrologic/hydraulic information and for sure offers the possibility of interesting discussions.

The manuscript is well written and organized; the methodology is almost well described, even if additional details could be included to help for reader understanding. Summarizing, the topic is of interest for the scientific community and the manuscript could be eventually considered for publication in this Journal after some issue are addressed by the Authors. My comments about the work are listed in the following paragraph; I hope that they will be helpful for manuscript improvement.

With appreciation,

Elena Volpi

**Specific comments**

1. The Authors introduce the bootstrap approach (l. 5-9 p. 3) as a solution to overcome the problem of isolated historical events for which confidence intervals are typically not symmetrical. It is not clear what the Authors mean by symmetrical confidence intervals; this issue should be explained since it is the motivation (together with the easy application of FFA) for reconstructing a continuous data set. Further, bootstrap is not necessary for confidence interval estimation (l. 9-10 p. 3) yet still necessary for continuous data set reconstruction.

2. The hydraulic model is used to propagate the discharge for the historic flood events reconstructed by Meurs (2006) from Cologne to Lobith; to this aim the Authors state that they use the current geometry of the riverbed and floodplain in order to correct the historic floods for anthropogenic interventions and natural changes of the river system, which is referred as "normalization" in the manuscript (l. 10-14 p.3). This approach is unusual based on my experience

(Calenda et al., 2005); historical flood events should be simulated by reconstructing the historical conditions (the river geometry as in the period the flood occur), that is what Authors would have available if measures would have started in the ancient past. In essence, I am not convinced that propagating the ancient floods in the current riverbed is the correct approach to solve the "homogenization" problem; conversely, this "gives insight in the consequences of an event with the same characteristics of a historic flood event translated to present times" (as stated by the Authors themselves at l. 17-18, p. 3).

3. Based on my opinion the Authors should "naturalize" the estimated discharge, by computing the discharge that they would have observed in absence of some anthropogenic change in the riverbed or in the catchment (l. 14-16 p.3). This means that are the recent events that should be reported to pre-dike conditions and not the opposite (as done in Section 2.3.2). The presence of the dike artificially alters the natural regime of the extreme flood events; the anthropogenic alteration of flood regime should be of deterministic nature, even if its estimation is characterized by a certain degree of uncertainty.

4. Why do the normalized events almost always lead to a higher discharge than the historic event (l. 16-17, p. 3)?

5. Section 2. For the sake of clarity, a table summarizing the type of information and the related uncertainty for the different time periods should be included.

6. L. 14-15, p. 4. The Authors should clarify the distance and the characteristics of the nearby gauging locations.

7. The procedure discussed in Section 3 is based on a non-parametric approach; alternatively a parametric method, based on the same assumption that ancient flood events follow the same statistical behavior of those systematically recorded, could have been considered. See Stedinger and Cohn (1986) and Francés

(1998); for an application to a case study similar to that discussed in this paper see also Calenda et al. (2009). Which is the advantage of the approach used in this work?

8. L. 2-7, p. 9. The Authors states that "the available goodness-of-fit tests for selecting an appropriate distribution function are often inconclusive. Those tests are more appropriate for the central part of the distribution than for the tail (Chbab et al., 2006), where we are interested in since the tail determines the investments required for future flood protection measures." I agree with the Authors that goodness-of-fit tests might be inconclusive, as discussed deeply in Serinaldi et al. (2018); on the other hand they provide a first indication on which models, among several competing ones, could be excluded due to the poor performance (see, e.g., Laio, 2004). In such a sense, I suggest the Authors at least to rephrase the sentence, also because there are different goodness-of-fit test which focus on the statistical behavior of the tails, such as the Anderson-Darling test and the Modified Anderson-darling test (Laio, 2004).

9. Following the argument of previous comment, I do not believe that restricting the analysis to a single probability distribution model (although it is the Generalized Extreme Value distribution commonly used in literature to perform an FFA) is a good choice. Since the interest is in evaluating how the confidence bounds of extreme quantile estimates reduce when adding the historical information (l. 18-21 p. 9), it should be considered that confidence bounds depend not only on the length and information content of the dataset but also on the probability model itself. Hence, results could be different if a different model is taken into account.

10. L. 10-12 p. 9. Do you the Authors mean that they assume an upper bounded distribution? This issue should be clarified.

11. Figure 5 is unnecessary, It could be removed.

12. Figure 6. The largest extreme events are not included in the uncertainty bounds. The corresponding sample bounds could be included as well to text the model performance (see comment 9).

13. Section 5.2. I am not sure I fully understood the rationale and the approach behind the analysis performed here. The historical events are some of the highest events observed in the whole observation period. If a sample is reconstructed by simply resampling the events observed in 1901-2018 (without including the largest historical events but with the same length of that used in previous sections), the largest events might only be those observed in the more recent period; as a consequence, the fitted model is expected to be characterized by, e.g., a smaller variance, which implies narrower uncertainty bounds. I do not see this behavior in figure 7 (upper panel). What I see in figure 7 is that the fitted model in the two cases is almost the same, while the uncertainty bounds are significantly different. I can explain this only if the reconstructed samples have a very different length. Please provide a deeper explanation.

14. L. 20-22 p. 14. It is not clear how the extended data set with normalized reconstructed discharges can capture the long-term climatic variability (see also previous comments).

15. L. 35, p. 14. Isn't it the 1374 event?

16. Fig. 8. Adding one event equal to the largest one over a record is expected to affect somewhat the estimated model if the record is 100 years while non changes in the model are expected if the record is about 700 years. Hence, which is the lesson learned from this analysis?

17. Within the Conclusion Section a detailed list of the limitations of the approach proposed here should be provided.

**References**

- Calenda, G., Mancini, C. P., Volpi, E. (2005). Distribution of the extreme peak floods of the Tiber River from the XV century. Advances in Water Resources, 28(6), 615-625.

- Calenda, G., Mancini, C. P., Volpi, E. (2009). Selection of the probabilistic model of extreme floods: The case of the River Tiber in Rome. Journal of Hydrology, 371(1-4), 1-11.

- Francés, F., 1998. Using the TCEV distribution function with systematic and non-systematic data in a regional flood frequency analysis. Stochastic Hydrology and Hydraulics 12, 267–283.

- Laio, F., 2004. Cramer-von Mises and Anderson–Darling goodness of fit tests for extreme value distributions with unknown parameters. Water Resources Research 40, WR09308.

- Serinaldi, F., Kilsby, C. G., Lombardo, F. (2018). Untenable nonstationarity: An assessment of the fitness for purpose of trend tests in hydrology. Advances in Water Resources, 111, 132-155.

- Stedinger, J.R., Cohn, T.A., 1986. Flood frequency analysis with historical and paleoflood information. Water Resources Research 22 (5), 785–793.

---

## Referee Comment (RC2) · Anonymous Referee #2 · 12 May 2019

This paper discuss the extension of a flood series, based on hydraulic modelling and the utilisation of extend hydrological time series to estimate the frequency of extreme floods at the Rhine gauge Lobith. The authors expect a reduced sampling effect. It is widely known that for extreme events the empirical exceedance probabilities in short observation series are often overestimated. To solve this problem the authors suggest to extend the observed time series. In their case study they propose to extent the existing series of observations between 1901-2018 by a linear regression of water levels with neighbouring gauges for the period 1772 to 1900 based on a previous study from Toonen (Toonen, 2015) and the translation of these water levels into discharges using a stage-discharge relationship, which is not specified in detail. The resulting series

(1772-2018) is named as the "systematic" time period. The other and even more uncertain step consists in an estimation of the peaks of historic floods at Lobith. Here a series of 12 historic flood events in Cologne since 1342, provided by Meures and Herget is used. As these events were estimated more than 150 km upstreams, a (1D-2D) coupled hydraulic model is used to transfer these peaks to Lobith: "The reconstructed maximum discharges at Cologne (Meurs, 2006), which are not normalized for anthropogenic interventions upstream of Cologne, are used to predict maximum discharges at Lobith with the use of a hydraulic model to normalize the data set." The meaning of "normalization" in this context stays unclear. It seems to be the adaptation of these peaks (which were roughly estimated by Meures) on to today's conditions. There are extreme uncertainties connected with this approach: the river reach changed in its hydraulic characteristics over 700 years, the water levels in Cologne dating back several hundreds of years are uncertain, the discharges as well and so on. It is a big surprise that the authors are able to specify in Fig. 3 95% confidence intervals for the maximum discharges in Cologne and Lobith for these 12 events. It stays unclear how these intervals were estimated. The authors propose a bootstrap sampling method to fill the gaps between the historic floods with annual flood peaks from the systematic data set, that have an expected value lower than the sampled perception threshold which is set as the smallest flood among the historic peaks. This is approach seems to be critical as it does not add any information to the statistical analysis. The todays conditions are modified by the first extension to the part of the series until 1772. With the sampling the authors accept that the flood series consist of independent and identically distributed random variables, which is not certain. By definition bootstrapping is any test or metric that relies on random sampling with replacement. Here the wording "resampling of the non-systematic time series below the perception threshold" would be more appropriated. This has been done 5000 times and also the historical floods are varied within their 95% confidence intervals (however these were estimated!). The systematic series were not changed. The GEV was estimated for each of these samples, the distributions were averaged (!) and their 95% percent confidence bounds were estimated.

Table 1 specifies these 95% bounds with the 2-sigma-reach, this would be only justified if the quantiles would be normal distributed. I suppose that this is not the case. In total the value of this resampling study stays unclear for me as it does not extend the information content. The information, derived from the systematic series are used in a simulation study, but the basic assumption that the floods between 1772 and 1900 are reconstructed correctly adds uncertainty to it. There are at least two other options to consider historic floods in statistics:

REIS D. S., JR.; STEDINGER J. R. (2005): Bayesian MCMC flood frequency analysis with historical information. In: Journal of Hydrology, 313, pp. 97–116 (cited by the authors)

Wang, Q. J. (1990): Unbiased estimation of probability weighted moments and partial probability weighted moments from systematic and historical flood information and their application to estimating the GEV distribution. In: Journal of Hydrology 120 (1-4), S. 115–124

Both methods combine the information from the systematic data with istoric floods without assumption that these observations are representative for the historic series. In both methods, it is assumed that the historic floods are representative for today's conditions. These events are used to improve the estimation of the upper tail only. The systematic part of the series stays untouched. In this way the uncertainty of assumptions of a large part of the time series is avoided. The statement of the authors: "Most studies found that the confidence intervals of design discharges were reduced significantly by extending the systematic data set with historic events." does not mean that an artificially extended systematic dataset would be beneficial if it was expanded with uncertain assumptions about past flood conditions and their adaptation to the current situation.

My summary: The manuscript has some weakness with regard to uncertainty assessments (confidence intervals) where the methodology is not sufficient described. The

assumption of a symmetrical interval seems to be arbitrarily. Nevertheless the topic is interesting, the manuscript should be consider the existing state of the art in this field and compare its results with well-established existing methods.

I suggest to reject the manuscript for major revisions.

---

## Author Comment (AC1) · 13 May 2019

**[General reply from the authors]**

We would like to thank the reviewer for taking the time to review our manuscript. We highly appreciate her suggestions and comments, which are helpful in improving the manuscript. Below we have replied to the various comments made by the reviewer.

**[Replies to reviewer comments]**

1. The Authors introduce the bootstrap approach (l. 5-9 p. 3) as a solution to overcome the problem of isolated historical events for which confidence intervals are typically not symmetrical. It is not clear what the Authors mean by symmetrical confidence intervals; this issue should be explained since it is the motivation (together with the easy application of FFA) for reconstructing a continuous data set.

Symmetrical means that the confidence intervals follow a normal distribution. Hence, the 95% confidence intervals can be computed with the basic rule of +/- 1.96*standard deviation. However, confidence intervals are typically not symmetrical for flood frequency relations. Hence, these intervals are difficult to compute if the data of annual maximum discharges is extended with historic events in isolation. Therefore, we would like to create a continuous data set such that the method to compute confidence intervals remains unchanged compared to traditional FFAs. Please also see the next comment.

Further, bootstrap is not necessary for confidence interval estimation (l. 9-10 p. 3) yet still necessary for continuous data set reconstruction.

It is indeed true that a bootstrap approach is not needed to compute the confidence intervals if a continuous data set is present. However, a bootstrap method is still needed to create a continuous data set as was done in this study. Both are different kind of bootstrap approaches. Using the same terminology leads to confusion. Therefore, the section will be adapted to (with in green the new sentences):

"The objective is to develop a straightforward method to consider historic flood events in an FFA, while the basic principles of an FFA remain unchanged. Confidence intervals of flood frequency relations are typically not symmetrical distributed (Schendel and Thongwichian, 2017). This means that the confidence intervals do not obey a normal distribution, but they are skewed. For a continuous data set, the asymmetric distributed confidence intervals can be computed relatively easily, while this becomes more problematic if historic flood events are added to the data set of measured discharges in isolation. To overcome this problem, bootstrap approaches such as the test inversion bootstrap method are recently developed (e.g. Burn (2003); Kyselý (2008); Schendel and Thongwichian (2017)). This study is novel since a continuous data set is created. The use of a bootstrap approach to compute the confidence intervals is now redundant. Although still a bootstrap approach is required to create the continuous data set, the method to compute the confidence intervals does not change compared to an FFA solely based on measured annual maximum discharges. This makes the comparison between the confidence intervals of the measured annual maximum discharges and the extended data set more reasonable."

2. The hydraulic model is used to propagate the discharge for the historic flood events reconstructed by Meurs (2006) from Cologne to Lobith; to this aim the Authors state that they use the current geometry of the riverbed and floodplain in order to correct the historic floods for anthropogenic interventions and natural changes of the river system, which is referred as "normalization" in the manuscript (l. 10-14 p.3). This approach is unusual based on my experience (Calenda et al., 2005); historical flood events should be simulated by reconstructing the historical conditions (the river geometry as in the period the flood occur), that is what Authors would have available if measures would have started in the ancient past. In essence, I am not convinced that propagating the ancient floods in the current riverbed is the correct approach to solve the "homogenization" problem; conversely, this "gives insight in the consequences of an event with the same characteristics of a historic flood event translated to present times" (as stated by the Authors themselves at l. 17-18, p. 3).

It is indeed true that historic flood events should be reconstructed based on the historical conditions. This is exactly what Meurs (2006) has done. Historic flood events were reconstructed near the city of Cologne, Germany, based on reconstructed main channel bathymetry.

However, our aim in this paper was not to make reconstructions of the historic events along the river stretch. In this paper, we aimed to predict flood frequency relations for current water policy assessments and therefore we would like to have the present-day discharges. This is why 'normalization' is done in the Dutch water policy. Even the measured discharges in e.g. 1920 are normalized to present-day discharges since the river system has altered a lot due to human interventions resulting in a change of the flood frequency relation. Nowadays, more water is capable of flowing through the river system towards Lobith, German-Dutch border, as a result of the heightened dikes along the Lower Rhine. Therefore, the historic flood events have no predictive value without normalzing it into present-day discharges. This is why we have normalized the historic flood events at Cologne, which are based on historical information, to present-day discharges at Lobith. To do so, we use the hydraulic model which is based on the current geometry. This hydraulic model is described in Bomers et al. (2019), and now accepted for publication in Natural Hazards.

3. Based on my opinion the Authors should "naturalize" the estimated discharge, by computing the discharge that they would have observed in absence of some anthropogenic change in the riverbed or in the catchment (l. 14-16 p.3). This means that are the recent events that should be reported to pre-dike conditions and not the opposite (as done in Section 2.3.2). The presence of the dike artificially alters the natural regime of the extreme flood events; the anthropogenic alteration of flood regime should be of deterministic nature, even if its estimation is characterized by a certain degree of uncertainty.

For flood safety assessments, we are interested in the current flooding regime and not that of the pre-dike conditions. It is indeed true that the presence of the dike alters the natural regime of the extreme flood events, but we are interested in this change since it determines how much water can enter the Netherlands at Lobith nowadays. Therefore, normalization of the historic flood events to present-day conditions is of high importance to correctly estimate flood frequency relations of the present river system.

4. Why do the normalized events almost always lead to a higher discharge than the historic event (l. 16-17, p. 3)?

This is because more water is capable of flowing through the river system as a result of the heightened dikes along the Lower Rhine. Nowadays, floods occur for higher discharge stages compared to the historical time period. This will be added in the revised manuscript

5. Section 2. For the sake of clarity, a table summarizing the type of information and the related uncertainty for the different time periods should be included.

A table with the various types of uncertainties for each time period will be added in the revised manuscript. See table below.

| Time period | Data source | Cause uncertainty | Location |
|---|---|---|---|
| 1342-1771 | Meurs (2006) | Reconstruction uncertainty caused by uncertain main channel bathymetry, bed friction and maximum occurred water levels | Andernach |
| 1772-1865 | Toonen (2015) | Reconstruction uncertainty based on measured water levels of surrounding sites | Emmerich, Pannerden and Nijmegen |
| 1866-1900 | Toonen (2015) | Uncertainty caused by translation measured water levels into discharges | Lobith |
| 1901-1950 | Tijssen (2008) | Uncertainty caused by extrapolation techniques to translate measured velocities at the water surface into discharges | Lobith |
| 1950-2000 | Tijssen (2008) | Uncertainty caused by translation velocity-depth profiles into discharges | Lobith |
| 2000-2008 | Tijssen (2008) | Measurement errors for discharges slightly exceeding the bankfull discharge | Lobith |
| 2008-2018 | Measured water levels available at https://waterinfo.rws.nl | Measurement errors for discharges slightly exceeding the bankfull discharge | Lobith |

6. L. 14-15, p. 4. The Authors should clarify the distance and the characteristics of the nearby gauging locations.

The following will be added in the manuscript:

"For the period 1772-1865 water levels were measured at the nearby gauging locations Emmerich (Germany) located 10 kilometers in upstream direction, Pannerden located 10 kilometers in downstream direction and Nijmegen located 22 kilometers in downstream direction."

However, note that this analysis has been performed by Toonen (2015) and is not part of this paper. Therefore, we refer for more information about the characteristics of the 1772-1901 data set to Toonen (2015).

7. The procedure discussed in Section 3 is based on a non-parametric approach; alternatively a parametric method, based on the same assumption that ancient flood events follow the same statistical behavior of those systematically recorded, could have been considered. See Stedinger and Cohn (1986) and Francés

It is indeed true that a non-parametric approach could have been considered. However, in this paper we had the preference to create a continuous data set instead. This is because, since recently, the Dutch water policy uses a new method in which a continuous data set of 50,000 years based on resampled measured weather conditions (e.g. rainfall, temperature, evapotranspiration) is used to predict flood frequency relations (Hegnauer et al., 2014, and also described in Chbab (2006)). We wanted to use the method of Hegnauer et al. (2014) of creating a continuous data set to test whether it also works with resampling measured discharges. This makes the use of HBV and Hydraulic modelling to translate the weather data into maximum discharges redundant, as was done by Hegnauer et al. (2014).

Furthermore, we wanted to create a continuous data set since the computation of the confidence intervals of a flood frequency relation remains unchanged compared to the analysis of just measured annual maximum discharges, making the comparison between the two more reasonable. This argument will be added in the introduction of the revised manuscript. For future work, it is interesting to study how confidence intervals deviate between the proposed methodology and a method based on a parametric approach. However, our results are in line with the findings of Francés (1998), who also showed that the uncertainty intervals of FFAs reduces if historical information is included in the analysis.

8. L. 2-7, p. 9. The Authors states that "the available goodness-of-fit tests for selecting an appropriate distribution function are often inconclusive. Those tests are more appropriate for the central part of the distribution than for the tail (Chbab et al., 2006), where we are interested in since the tail determines the investments required for future flood protection measures." I agree with the Authors that goodness-of-fit tests might be inconclusive, as discussed deeply in Serinaldi et al. (2018); on the other hand they provide a first indication on which models, among several competing ones, could be excluded due to the poor performance (see, e.g., Laio, 2004). In such a sense, I suggest the Authors at least to rephrase the sentence, also because there are different goodness-of-fit test which focus on the statistical behavior of the tails, such as the Anderson-Darling test and the Modified Anderson-darling test (Laio, 2004).

We agree with you that there are various goodness-of-fit tests, all with their own properties. The sentence will be rewritten with in green the new text:

"A probability distribution function is used to fit the annual maximum discharges to its probability of occurrence. Many types of distribution functions and goodness-of-fit tests exist, all with their own properties and drawbacks. However, the available goodness-of-fit tests for selecting an appropriate distribution function are often inconclusive. This is mainly because each test is more appropriate for a specific part of the distribution, while we are interested in the overall fit of the distribution. This is because the safety standards expressed in probability of flooding along the Dutch dikes vary from $10^{-2}$ to $10^{-5}$. We restrict our analysis to the use of a Generalized Extreme Value (GEV) distribution since this is commonly used in literature to perform an FFA"

9. Following the argument of previous comment, I do not believe that restricting the analysis to a single probability distribution model (although it is the Generalized Extreme Value distribution commonly used in literature to perform an FFA) is a good choice. Since the interest is in evaluating how the confidence bounds of extreme quantile estimates reduce when adding the historical information (l. 18- 21 p. 9), it should be considered that confidence bounds depend not only on the length and information content of the dataset but also on the probability model itself. Hence, results could be different if a different model is taken into account.

You are indeed correct that the uncertainty interval also highly depends on the fitted distribution itself. Although not shown, we performed the analysis with other distributions as well (e.g. Weibull and Gumbel) and the general conclusion of 'reduction of the confidence bounds as a result of extending the data set of measured discharges' also holds for these distributions. For the GEV distribution we found a reduction of 73% as a result of extending the data set of annual measured discharges with historic events, with the Gumbel distribution a reduction of 60% and with the Weibull distribution a reduction of 67%.

We will add a sentence to the revised manuscript in which it is stated that also for other distribution functions a reduction of the confidence interval was found. However, we will not show the in-depth results of different distribution types, because we think this is distracting the reader from the analysis performed and corresponding main findings. Furthermore, the GEV distribution has been shown to fit the data of the Rhine river well and therefore this distribution was preferred above other distributions. Finally, we would like to highlight that many closely-related studies also only focused on the use of a single distribution (e.g. Francés (1998)).

10. L. 10-12 p. 9. Do you the Authors mean that they assume an upper bounded distribution? This issue should be clarified.

Yes, we indeed assume an upper bounded distribution. The GEV distribution has an upper bound as a result of the shape parameter which both influences the skewness and kurtosis of the distribution. We use a bounded distribution since the maximum discharge that is capable of entering the Netherlands at Lobith is limited to a physical maximum value. The crest levels of the dikes along the Lower Rhine are not infinitely high. The height of the dikes influences the discharge capacity of the Lower Rhine and hence the discharge that can flow towards Lobith. This explanation will be added to the revised manuscript such that it becomes clear why we use an upper bounded distribution.

11. Figure 5 is unnecessary, It could be removed.

Figure 5 will be removed from the revised manuscript.

12. Figure 6. The largest extreme events are not included in the uncertainty bounds. The corresponding sample bounds could be included as well to text the model performance (see comment 9).

Also the largest extreme events are included in the uncertainty bounds (see table 1). However, since the upper bound of the measured data set has a value of 29,631 m3/s (table 1) this line was not entirely drawn. Since it leads to confusion, we will plot the entire line in the revised manuscript. See the figure below.

[Figure]

13. Section 5.2. I am not sure I fully understood the rationale and the approach behind the analysis performed here. The historical events are some of the highest events observed in the whole observation period. If a sample is reconstructed by simply resampling the events observed in 1901-2018 (without including the largest historical events but with the same length of that used in previous sections), the largest events might only be those observed in the more recent period; as a consequence, the fitted model is expected to be characterized by, e.g., a smaller variance, which implies narrower uncertainty bounds. I do not see this behavior in figure 7 (upper panel). What I see in figure 7 is that the fitted model in the two cases is almost the same, while the uncertainty bounds are significantly different. I can explain this only if the reconstructed samples have a very different length. Please provide a deeper explanation.

You are indeed correct that we simply resample the events observed in 1901-2018 without including the largest historical events but with the same length. This corresponds with the line 'Bootstrap 1901-2018 data'. This data set has a length equal to the 1317-2018 period. If we compare the line with the Bootstrap 1317-2018 data set, we indeed see that the uncertainty interval of the Bootstrap 1901-2018 is still larger even though the length of the two data sets are the same. It must be noted that not only the length influences the uncertainty interval, but also the discharges within the data set and resulting variance.

For the Bootstrap 1901-2018 data set, the entire measured data set (1901-2018) is used for resampling. The created continuous series (5,000 in total for convergence reasons) has an average variance of 4,19 x 10$^6$ m$^3$/s. For the Bootstrap 1772-2018 data set, only the discharges below a certain threshold in the measured time period (1772-2018) are used for resampling. In this study, the perception threshold was chosen to be equal to the lowest flood event in the historical time period having a discharge of between 6,928-10,724 m$^3$/s. Hence, the missing years in the historical time

period are filled with relatively low discharges, but some of the largest events in the historical time period are larger than ever measured. The total variance of the data set decreases (3.35 x 10$^6$ m$^3$/s) as a result of the lower discharges to create the continuous data set. As a result of the lower variance, also the uncertainty bounds are smaller compared to the Bootstrap 1901-2018 data set. This explanation will be added in the revised manuscript.

14. L. 20-22 p. 14. It is not clear how the extended data set with normalized reconstructed discharges can capture the long-term climatic variability (see also previous comments).

The historic flood events are only normalized for changes in the river system. As a result, the normalized discharges still capture the climatic conditions in the historical time period. Although the missing years within the historical time period are filled with the measured data set 1772-2018, the most extreme events still capture the climatic variability in the period ~1300-2018. This will be added in the revised manuscript.

15. L. 35, p. 14. Isn't it the 1374 event?

The 1374 flood event is indeed the largest observed discharge (at Cologne) of the last 1,000 years. However, in this analysis we consider the largest measured discharge (measurements have been performed since 1900), which correspond with the 1926 flood event.

16. Fig. 8. Adding one event equal to the largest one over a record is expected to affect somewhat the estimated model if the record is 100 years while non changes in the model are expected if the record is about 700 years. Hence, which is the lesson learned from this analysis?

The lesson learned is that flood safety assessments become more robust if the data set of annual maximum discharges is extended. After the 1993 and 1995 flood events of the Rhine river, the flood frequency relation altered significantly resulting in an increase of the design discharge at Lobith of 1,000 m3/s. Such an increase in the design discharge requires huge investments to cope with the new flood safety standards which were set after the 1993 and 1995 floods. Such an increase was not found if a longer time series was included in the analysis. Looking at the results, decision makers might have taken a different decision.

17. Within the Conclusion Section a detailed list of the limitations of the approach proposed here should be provided.

A list of the limitations of the proposed method will be included in the discussion which is:

- The 1772-2018 measured data set is used to create a continuous data set. Preferably, we would like to have a historical continuous record since now the low flows (discharges with high probability of occurrence) are biased on climate conditions of the last 250 years
- Historical flood events must be normalized for anthropogenic and natural changes in the river system which can be quite time demanding in terms of computational time
- The extended data set still has limited length. Preferable we would like to have a data set of e.g. 100,000 years such that extrapolation to such return periods is not required anymore. However, the proposed method is a large step to decrease uncertainty.

- The predicted uncertainty intervals depend on the chosen perception threshold. A larger threshold results in an increase of the variance of the data set and hence to an increase in the uncertainty intervals.
- The shape of the constructed FF curve strongly depends on the climate conditions of the period considered. If the data set is extended with a period which only has a small number of large flood events, this will result in a significant shift of the FF curve in downward direction. This shift can be overestimated if the absence of large flood events only applies to the period used to extend the data set.

Up till now, only the last point was mentioned in the discussion. The other points will be added.

References

Bomers, A., Schielen, R.M.J., Hulscher, S.J.M.H., 2019. Consequences of dike breaches and dike overflow in a bifurcating river system. Accepted for publication in: *Natural Hazards.*

Hegnauer, M., Beersma, J.J., van den Boogaard, H.F.P., Buishand, T.A., Passchier, R.H., 2014.Generator of Rainfall and Discharge Extremes (GRADE) for the Rhine and Meuse basins. Final report of GRADE 2.0. Technical Report. Deltares. Delft, The Netherlands

---

## Referee Comment (RC3) · Anonymous Referee #3 · 21 May 2019

In this paper, the authors present a method/case study to reconstruct a continuous times series of annual maximum discharges in order to estimate return times for flood discharges for the Rhine at Lobith. The study uses modern data from 1901 onwards, discharges reconstructed from water level measurements back to 1772 and information from historical flood events back to the 1300. Extending a time series with this information leads to a reduction of uncertainty and to more stable return times. The paper is well structured and written, and the topic is of relevance for flood risk estimation. However, there are some major issues which need to be addressed before this paper can be published from my point of view:

[Figure]

One general problem I have with this manuscript is that the authors refer to and use data from many other studies, especially the one from Toonen (2015). It is difficult to follow the article for reader if one is not familiar with these studies because it requires reading many secondary sources to gain insight on how all the different data(-sets) were collected and obtained, e.g. how was the regression analysis by Toonen (2015) performed, how were the historical floods in Cologne by Herget and Meurs (2010) reconstructed, etc.

This paper includes a lot of different data sets (systematic, historical, plus various bootstrapped time series), it would be beneficial for readers to include a table with a short description and overview of the properties of these data sets and to name them consistently throughout the paper.

The term "normalize" is used in different contexts (e.g. for historical floods, for the 1900-2008 data set, for the data set of Toonen (2015) which is not normalized but used as normalized data). I find this confusing since it does not become clear what is actually meant by this and what has been done to "normalize" each of these data sets. A more thorough explanation on this matter would be useful.

In section 2.2 the authors describe the Toonen (2015) data set which uses a linear regression to compute water levels at Lobith. This method leads to a reduced variance of this data set (c.f. table 1). How would this affect the bootstrapping later on, if samples from the so called "systematic time period" with different variances (1772-1900, 1901-2018) are drawn?

From my point of view, the section 2.3.2 presenting the normalization of historical flood events leaves some open questions which need to be addressed. Using a coupled 1D/2D model to route the discharges from Cologne to Lobith seems a reasonable approach given the circumstances of the data, but the dike breach model and the underlying assumptions need more explanation. Is it valid to assume dike breach parameters from today's river geometry for historical times? Is there any historical evidence

that there were dike breaches in the past, especially the 1374 event? Especially the reduction of the 1374 flood peak from Cologne to Lobith needs some sound justification/explanation. Why is this reduction only occurring for this specific event? Were there also dike breeches for the other historical events? What exactly is meant by "the upstream discharge shape is varied" (p.6, line 12)? There is a lot of uncertainty in this, which somehow contradicts the aim of the paper to reduce uncertainty. Furthermore, it would be interesting to know if any of 12 historical flood events where winter events, where ice draft/ice jams could/did play a role. Furthermore, assuming a normal distribution of uncertainties is valid for discharge measurements, but is this also the case for the estimation of historical extreme floods? Or is any discharge values in the uncertainty range equally possible? The reconstruction of the events in Cologne is based on the Manning equation and the uncertainty range results from different roughness coefficients. But do all of these follow a normal distribution?

Section 3: The bootstrap method to create continuous times series is a reasonable approach, however it would also be possible, to use the maximum likelihood method and incorporate the uncertainty range of the historical discharges as well as the discharges lower the perception threshold in the parameter estimation. From my point of view this approach is straight forward and should yield similar results. Could the authors explain/discuss the benefit of the bootstrapping approach?

In Section 4, the authors state that there are many distributions and fitting methods for flood frequency analysis and that the only use the GEV with maximum likelihood method. It seems justified, that only one combination is used to quantify the reduction of the uncertainty, but in practice there are many different distributions and parameter estimation methods - which again cause higher uncertainties in the estimation of return times, especially for the upper tail extremes. The authors should include a comment and if possible a quantification of this effect on this in the discussion. In Section 5.2., the authors argue that the reconstruction of historical flood events is complicated and time consuming and that this can be overcome by bootstrapping. However, the information

from rare and large historical flood events is still required as is stated at the end of the section. This sounds like an inconsistency in the line of argumentation. Furthermore, this whole section is somewhere between results and discussion. I suggest that the authors try to separate more clearly between results and discussion.

In the discussion, the effect of a hypothetical future extreme flood on the robustness of return times is addressed, which is somehow obvious from my point of view. This aspect does not add much value to the paper and can either be omitted or be moved to the results section.

Some specific comments:

Page 3, line 3f.: Why are uncertainties not symmetrical due to missing continuous data? Don't these result from the non-linearity of the rating curves?

Page 4, line 7f.: ACDP-measurements are in general not free of uncertainties, this assumption is not correct.

Page 11, line 2: Where does this confidence interval of $7400 m^3/s$ come from?

Page 15, line 1: Same as above, modern discharge measurements are not free of measurement errors!

Page 15, line 5f.: See above, this is not a novel results and can more or less be expected. Furthermore, the statement that "flood managers can be less nervous" sounds awkward and is not really correct, since the uncertainty caused by different distributions/parameter estimation methods is not addressed.

Figure 2: Should be replaced by a "conventional" map, including national boundaries, a scale bar etc. Readers from outside of Europe might not be familiar with this region.

Table 1: The results of Toonen (2015) can be omitted in this table from my point of view.

Figure 6 and 7: The colours/line styles of the different curves are difficult to distinguish

and should be changed to make these figures better to read.

References: To my knowledge, Meurs 2006 is a diploma thesis, not a PhD thesis.
* * *

---

## Author Comment (AC2) · 14 Jun 2019

**[General reply from the authors]**

We would like to thank the anonymous reviewer for taking the time to review our manuscript. We highly appreciate the suggestions and comments, which are helpful in improving the manuscript. Below we have replied to the various comments made by the reviewer.

**[Replies to reviewer comments]**

This paper discuss the extension of a flood series, based on hydraulic modelling and the utilisation of extend hydrological time series to estimate the frequency of extreme floods at the Rhine gauge Lobith. The authors expect a reduced sampling effect. It is widely known that for extreme events the empirical exceedance probabilities in short observation series are often overestimated. To solve this problem the authors suggest to extend the observed time series. In their case study they propose to extent the existing series of observations between 1901-2018 by a linear regression of water levels with neighbouring gauges for the period 1772 to 1900 based on a previous study from Toonen (Toonen, 2015) and the translation of these water levels into discharges using a stage-discharge relationship, which is not specified in detail.

It is indeed true that the translation of the water levels into discharges for the period 1772-1900 was not specified in detail. This is because this has been described in detail by Toonen (2015) who performed the analysis. To help the readers of our manuscript, the following will be added in the revised manuscript in section 2.2, with in green the new text:

"For the period 1772-1900, the data presented by Toonen (2015) is used. At Lobith, daily water level measurements are available since 1866. For the period 1772-1865 water levels were measured at the nearby gauging locations Emmerich, Pannerden and Nijmegen. Toonen (2015) used the water levels of these locations to compute the water level at Lobith and associated uncertainty interval with the use of linear regression between the different measurement locations. Subsequently, he translated these water levels, together with the measured water levels for the period 1866-1900, into discharges using stage-discharge relations at Lobith. These relations were derived based on discharge predictions adopted from Cologne before 1900 and measured discharges at Lobith after 1900, and water levels estimates from the measurement locations Emmerich, Pannerden, Nijmegen and Lobith. Since the discharge at Cologne strongly correlates with the discharge at Lobith, the measured discharges in the period 1817-1900 could be used to predict discharges at Lobith. Hence, the reconstructed water levels were used to derive stage-discharge relations. The 95% confidence interval in reconstructed water levels propagates in the application of stage-discharge relations, resulting in an uncertainty range of approximately 12% for the reconstructed discharges (Fig. 1). The reconstructed discharges in the period 1772-1900 represent the computed maximum discharges at the time of occurrence and has not been normalized for changes in the river system."

The resulting series (1772-2018) is named as the "systematic" time period. The other and even more uncertain step consists in an estimation of the peaks of historic floods at Lobith. Here a series of 12 historic flood events in Cologne since 1342, provided by Meures and Herget is used. As these events were estimated more than 150 km upstreams, a (1D-2D) coupled hydraulic model is used to transfer these peaks to Lobith: "The reconstructed maximum discharges at Cologne (Meurs, 2006), which are not normalized for anthropogenic interventions upstream of Cologne, are used to predict maximum discharges at Lobith with the use of a hydraulic model to normalize the data set." The meaning of

"normalization" in this context stays unclear. It seems to be the adaptation of these peaks (which were roughly estimated by Meures) on to today's conditions.

We indeed mean with normalization adapting the historic peaks at Cologne on today's geometry conditions. Hence we will find the maximum discharges at Lobith as a result of the maximum discharges at Cologne under current river conditions. Please see page 3, lines 13-14: "In such a way, the historic floods are corrected for anthropogenic interventions and natural changes of the river system, referred to as normalization in this study."

There are extreme uncertainties connected with this approach: the river reach changed in its hydraulic characteristics over 700 years, the water levels in Cologne dating back several hundreds of years are uncertain, the discharges as well and so on. It is a big surprise that the authors are able to specify in Fig. 3 95% confidence intervals for the maximum discharges in Cologne and Lobith for these 12 events. It stays unclear how these intervals were estimated.

The 95% confidence interval for the maximum discharges in Cologne were taken from Meurs (2006). His method is shown by Herget and Meurs (2010) in detail, using the 1374 flood event as a case study. The following will be added in the revised manuscript in section 2.3 with in green the new text:

"Meurs (2006) has reconstructed maximum discharges during historic flood events near the city of Cologne (Germany). The oldest event dates back to 1342. The used method is described in detail by Herget and Meurs (2010), in which the 1374 flood event was used as a case study. Historic documents providing information about the maximum water level during the flood event were combined with the reconstruction of the river cross section at that same time. Herget and Meurs (2010) calculated mean flow velocities near the city of Cologne at the time of the historic flood events with the use of the empirical Manning's equation:

$$Q_p = A_p R_p{}^{2/3} S^{1/2} n^{-1}$$

where $Q_p$ represents the peak discharge, $A_p$ the cross-sectional area during the highest flood level, $R_p$ the hydraulic radius during the highest flood level, S the slope and n the Manning's roughness coefficient.

However, the highest flood level as well as Manning's roughness coefficient are uncertain. The range of maximum water levels was based on historical sources, whereas the range of Manning's roughness coefficients were based on the tables of Chow (1959). With this information, Herget and Meurs (2010) were able to calculate maximum discharges of the specific historic flood events and associated uncertainty range (Fig. 3)."

The reconstructed historic discharges and their uncertainties were used as input data of the 1D-2D coupled model to compute resulting discharges at Lobith. This is a valid method since there is a strong correlation between the discharge at Cologne and Lobith for in channel flow conditions, even though Cologne is located roughly 160 km upstream of Lobith since they are located in the same fluvial trunk valley and only have minor tributaries (Sieg, Ruhr and Lippe) joining in between (Toonen, 2015). This will be added in the revised manuscript to clarify the applicability of using historical discharge reconstructions at Cologne to determine corresponding present-day maximum discharges at Lobith.

With the 1D-2D coupled model, a Monte Carlo analysis was performed for each historic flood event in which the following parameters were considered to be random: maximum upstream discharge (based on the uncertainty range of each historic flood event as reconstructed by Herget and Meurs (2010)), dike breach thresholds, dike breach formation time and final breach width. The method of this analysis is discussed in detail by Bomers et al. (2019) and has recently been accepted for publication. As a result of the uncertain upstream discharge and breach characteristics, also the discharge at Lobith for each historic event is uncertain. Therefore, many model runs are performed for each event until convergence in model results is reached. Hence, the expected discharge at Lobith and expected 95% confidence intervals were computed (Fig. 3). The hydraulic modelling approach to normalize the historic flood events will be explained in more detail in the revised manuscript.

The authors propose a bootstrap sampling method to fill the gaps between the historic floods with annual flood peaks from the systematic data set, that have an expected value lower than the sampled perception threshold which is set as the smallest flood among the historic peaks. This is approach seems to be critical as it does not add any information to the statistical analysis. The todays conditions are modified by the first extension to the part of the series until 1772. With the sampling the authors accept that the flood series consist of independent and identically distributed random variables, which is not certain.

Indeed, we assume independent and identically distributed random variables. The authors are aware of this assumption. However, please note that to perform a flood frequency analysis we always have to assume that the discharge observations are independent and stationary (Khaliq et al., 2006). Although the assumption is highly uncertain, it must be noted that up till now no consistent large-scale climate change signal in observed flood magnitudes has been identified (Blöschl et al., 2017) justifying the assumption of independent and identically distributed random variables.

By definition bootstrapping is any test or metric that relies on random sampling with replacement. Here the wording "resampling of the non-systematic time series below the perception threshold" would be more appropriated. This has been done 5000 times and also the historical floods are varied within their 95% confidence intervals (however these were estimated!). The systematic series were not changed.

Maybe this was not fully clear to the reviewer, but also the systematic data set was changed. For each year within the historical period of which no data is available, an annual maximum discharge of the systematic data set below the perception threshold was randomly drawn (See step 5 in Fig. 4). This corresponds with the bootstrap method. As a result, each created continuous data set is different.

Furthermore, the values within the systematic data set were varied within their 95% confidence intervals. The study described the uncertainties of the systematic data set, which vary for different time periods as a result of different measurement methods used. Please see Fig. 1 in the manuscript and Section 2.1. We will add a table in which all types of uncertainties are described for the various data sets used to extend the data set of maximum discharges.

The GEV was estimated for each of these samples, the distributions were averaged (!) and their 95% percent confidence bounds were estimated. Table 1 specifies these 95% bounds with the 2-sigmareach, this would be only justified if the quantiles would be normal distributed. I suppose that this is not the case.

We indeed, the confidence bounds of the discharges are not normally distributed. The caption of the table stating 2-sigma is not correct. If you look at the numbers of the uncertainty bounds you can already see that the confidence bounds are not normal distributed since the upper bound is much further away from the average value than the lower bound, specifically for a return period of 100,000 years. The caption will be changed accordingly in the revised manuscript.

In total the value of this resampling study stays unclear for me as it does not extend the information content. The information, derived from the systematic series are used in a simulation study, but the basic assumption that the floods between 1772 and 1900 are reconstructed correctly adds uncertainty to it.

We indeed assume that the 1772-1900 flood were reconstructed correctly, but not without uncertainty. We have included this uncertainty in the analysis. The 95% bounds of the 1772-1900 data set are determined by Toonen (2015) and explained in more detail above. He found an uncertainty interval of approximately 12%. This will be added explicitly to the revised manuscript to avoid further misunderstanding. Furthermore, we will add a table describing the uncertainties of the various data sets used to extend the systematic data set.

There are at least two other options to consider historic floods in statistics:

REIS D. S., JR.; STEDINGER J. R. (2005): Bayesian MCMC flood frequency analysis with historical information. In: Journal of Hydrology, 313, pp. 97–116 (cited by the authors)

Wang, Q. J. (1990): Unbiased estimation of probability weighted moments and partial probability weighted moments from systematic and historical flood information and their application to estimating the GEV distribution. In: Journal of Hydrology 120 (1-4), S. 115–124

Both methods combine the information from the systematic data with historic floods without assumption that these observations are representative for the historic series. In both methods, it is assumed that the historic floods are representative for today's conditions. These events are used to improve the estimation of the upper tail only. The systematic part of the series stays untouched. In this way the uncertainty of assumptions of a large part of the time series is avoided. The statement of the authors: "Most studies found that the confidence intervals of design discharges were reduced significantly by extending the systematic data set with historic events." does not mean that an artificially extended systematic dataset would be beneficial if it was expanded with uncertain assumptions about past flood conditions and their adaptation to the current situation.

We agree with you that we add uncertainty to the data set by adding historical flood events to the measured data set and by using a resampling method to create a continuous data set. However, it must be noted that many of the uncertainties of the historic flood events are included in the analysis, as well as the uncertainty of the systematic data set (1772-2018). The 95% confidence intervals of the flood frequency relations are thus based on these uncertainties.

It is true that our method influences the flood frequency curve in the domain of the systematic data set (discharges with high probability of occurrence). However, as far as we know this is always the case if the parameters of the (GEV) distribution are recomputed as a result of new data availability. If

we have a look at the figure below, we find that the design discharge with a return period of 100 years decreases from 12,080 m$^3$/s to 11,630 m$^3$/s by extending the systematic 1901-2018 data set towards 1317 using the bootstrap method. This decrease in design discharge corresponds with a change of 3.7% indicating that resampling the systematic data set of the historical time period only has a little effect on the shape of the flood frequency curve corresponding with high probability of occurrence. This justifies the use of the bootstrap method. Furthermore, we would like to highlight that we are typically interested in correct prediction of the tail, rather than the discharges with large probability of occurrence, since the tail (high return periods) is of high importance to design flood protection measures.

[Figure]

My summary: The manuscript has some weakness with regard to uncertainty assessments (confidence intervals) where the methodology is not sufficient described. The assumption of a symmetrical interval seems to be arbitrarily. Nevertheless the topic is interesting, the manuscript should be consider the existing state of the art in this field and compare its results with well-established existing methods. I suggest to reject the manuscript for major revisions.

We agree with the reviewer that we did not provide enough details about the considered uncertainties of the various data sets used. We will provide a detail explanation of the computed 95% confidence intervals of the following data sets in the revised manuscript:

- Reconstructed historic flood events at Cologne by Meurs (2006)
- Corresponding historic discharges at Lobith using a hydraulic model
- Reconstructed discharges for the period 1772-1900 by Toonen (2015)
- Measured discharges for the period 1901-2018

Furthermore, we will add more information about why we propose this method instead of a Bayesian method. We would like to highlight that our method is systematic. We can extend our data set with historical data and keep the method of a flood frequency analysis the same. In this way, we can make a clear comparison on the effect of extending the data set with multiple other sets on the confidence bounds of flood frequency analysis.

Although the maximum likelihood method only gives a point estimate of the (GEV) parameters, as sample size increases, maximum likelihood estimators become unbiased minimum variance estimators with approximate normal distributions. This is used to compute confidence bounds for

the GEV parameter estimates. We would like to highlight that, although the Bayesian method is capable of predicting parameter uncertainty without the assumption of being normally distributed, the results are influenced by the prior. The influence of the prior, which has to be defined by the modeler, on the posterior distribution of the parameters and hence on the uncertainty of flood frequency relations can even be larger than the influence of discharge measurement errors, as was found by Neppel et al. (2010). The disadvantage is thus that we have to choose the prior in the Bayesian method correctly such that the tail will be correctly predicted. However, we do not have any measurements in, or near to, the tail and consequently it is reasonable to estimate the prior by fitting the original data with the use of e.g. the Maximum Likelihood method. In this way, the benefits of the Bayesian method compared to a traditional flood frequency analysis are at least questionable.

We are aware that there is a strong debate between the 'Bayesians' and the 'Frequentist' in literature and discussion forums. With this paper, we do not want to get into this discussion. Rather, we wanted to show a novel and systematic approach which is easy to understand for practitioners to include historic flood information into flood safety assessments. The general methodology of a flood frequency analysis remains in this proposed bootstrap methodology, only the data set of measured discharges is extended. As a result, this method is close to current practice of water managers. We will add the reasons why we set up a bootstrap method in the revised manuscript and will compare the methodology with the Bayesian statistics.

REFERENCES:

Bomers, A., Schielen, R.M.J., Hulscher, S.J.M.H. (2019) Consequences of dike breaches and dike overflow in a bifurcating river system. Accepted for publication in: Natural Hazards. doi: 10.1007/s11069-019-03643-y.

Böschl, G., Hall, J., Parajka, J., Perdigão, R.A.P., Merz, B., et al. (2017) Changing climate shifts timing of European floods. In: Science 357, pp. 588–590. doi:10.1126/science.aan2506.

Frances, F. (1998) Using the TCEV distribution function with systematic and non-systematic data in a regional flood frequency analysis. In: Stochastic Hydrology and Hydraulics 12, pp. 267-283.

Khaliq, M.N., Ouarda, T.B., Ondo, J.C., Gachon, P., Bobée, B. (2006) Frequency analysis of a sequence of dependent and/or non-stationary hydro-meteorological observations: A review. In: Journal of Hydrology 329, pp. 534–552. doi:10.1016/j.jhydrol.2006.03.004

Neppel, L., Renard, B., Lang, M., Ayral, P.a., Coeur, D., Gaume, E., Jacob, N., Payrastre, O., Pobanz, K., Vinet, F.,(2010) Flood frequency analysis using historical data: accounting for random and systematic errors. In: Hydrological Sciences Journal 55, pp. 192–208. doi:10.1080/02626660903546092

---

## Author Response (AR1)

**Comments by Reviewer 1**

**[General reply from the authors]**

We would like to thank the reviewer for taking the time to review our manuscript. We highly appreciate her suggestions and comments, which are helpful in improving the manuscript. Below we have replied to the various comments made by the reviewer.

**[Replies to reviewer comments]**

1. The Authors introduce the bootstrap approach (l. 5-9 p. 3) as a solution to overcome the problem of isolated historical events for which confidence intervals are typically not symmetrical. It is not clear what the Authors mean by symmetrical confidence intervals; this issue should be explained since it is the motivation (together with the easy application of FFA) for reconstructing a continuous data set.

Symmetrical means that the confidence intervals follow a normal distribution. Hence, the 95% confidence intervals can be computed with the basic rule of +/- 1.96*standard deviation. However, confidence intervals are typically not symmetrical for flood frequency relations. Hence, these intervals are difficult to compute if the data of annual maximum discharges is extended with historic events in isolation. Therefore, we would like to create a continuous data set such that the method to compute confidence intervals remains unchanged compared to traditional FFAs. We removed the statement related to the symmetrical confidence intervals from the manuscript since we have revised the introduction. We now focus on the differences between Bayesian statistics and frequentist statistics and how both methods compute the confidence intervals of the parameters of the GEV distribution. See page 2 lines 4-11. The motivation of the study is now stated on page 2 lines 21-19.

Further, bootstrap is not necessary for confidence interval estimation (l. 9-10 p. 3) yet still necessary for continuous data set reconstruction.

It is indeed true that a bootstrap approach is not needed to compute the confidence intervals if a continuous data set is present. However, a bootstrap method is still needed to create a continuous data set as was done in this study. Both are different kind of bootstrap approaches. Using the same terminology leads to confusion. The statement about the bootstrap method to compute the confidence intervals is removed from the manuscript. Please also see the previous comment.

2. The hydraulic model is used to propagate the discharge for the historic flood events reconstructed by Meurs (2006) from Cologne to Lobith; to this aim the Authors state that they use the current geometry of the riverbed and floodplain in order to correct the historic floods for anthropogenic interventions and natural changes of the river system, which is referred as "normalization" in the manuscript (l. 10-14 p.3). This approach is unusual based on my experience (Calenda et al., 2005); historical flood events should be simulated by reconstructing the historical conditions (the river geometry as in the period the flood occur), that is what Authors would have available if measures would have started in the ancient past. In essence, I am not convinced that propagating the ancient floods in the current riverbed is the correct approach to solve the "homogenization" problem; conversely, this "gives insight in the consequences of an event with the same characteristics of a

historic flood event translated to present times" (as stated by the Authors themselves at l. 17-18, p. 3).

It is indeed true that historic flood events should be reconstructed based on the historical conditions. This is exactly what Meurs (2006) has done. Historic flood events were reconstructed near the city of Cologne, Germany, based on reconstructed main channel bathymetry. See page 5 lines 14-15.

However, our aim in this paper was not to make reconstructions of the historic events along the river stretch. In this paper, we aimed to predict flood frequency relations for current water policy assessments and therefore we would like to have the present-day discharges. This is why 'normalization' is done in the Dutch water policy. Even the measured discharges in e.g. 1920 are normalized to present-day discharges since the river system has altered a lot due to human interventions resulting in a change of the flood frequency relation. Nowadays, more water is capable of flowing through the river system towards Lobith, German-Dutch border, as a result of the heightened dikes along the Lower Rhine (see page 3 lines 2-4). Therefore, the historic flood events have no predictive value without normalizing it into present-day discharges. This is why we have normalized the historic flood events at Cologne, which are based on historical information, to present-day discharges at Lobith. To do so, we use the hydraulic model which is based on the current geometry. This hydraulic model is described in high detail in Bomers et al. (2019).

3. Based on my opinion the Authors should "naturalize" the estimated discharge, by computing the discharge that they would have observed in absence of some anthropogenic change in the riverbed or in the catchment (l. 14-16 p.3). This means that are the recent events that should be reported to pre-dike conditions and not the opposite (as done in Section 2.3.2). The presence of the dike artificially alters the natural regime of the extreme flood events; the anthropogenic alteration of flood regime should be of deterministic nature, even if its estimation is characterized by a certain degree of uncertainty.

For flood safety assessments, we are interested in the current flooding regime and not that of the pre-dike conditions. It is indeed true that the presence of the dike alters the natural regime of the extreme flood events, but we are interested in this change since it determines how much water can enter the Netherlands at Lobith nowadays. Therefore, normalization of the historic flood events to present-day conditions is of high importance to correctly estimate flood frequency relations of the present river system. Why normalization is of high importance is described on page 2 lines 34-35 and page 3 lines 1-2.

4. Why do the normalized events almost always lead to a higher discharge than the historic event (l. 16-17, p. 3)?

This is because more water is capable of flowing through the river system as a result of the heightened dikes along the Lower Rhine. Nowadays, floods occur for higher discharge stages compared to the historical time period. Please see page 3 lines 2-4.

5. Section 2. For the sake of clarity, a table summarizing the type of information and the related uncertainty for the different time periods should be included.

A table with the various types of uncertainties for each time period has been added to the revised manuscript. See table 1 on page 4.

6. L. 14-15, p. 4. The Authors should clarify the distance and the characteristics of the nearby gauging locations.

The following has been added in the manuscript:

"For the period 1772-1865 water levels were measured at the nearby gauging locations Emmerich (Germany) located 10 kilometers in upstream direction, Pannerden located 10 kilometers in downstream direction and Nijmegen located 22 kilometers in downstream direction." See page 4 lines 5-6.

However, note that this analysis has been performed by Toonen (2015) and is not part of this paper. Therefore, we refer for more information about the characteristics of the 1772-1901 data set to Toonen (2015).

7. The procedure discussed in Section 3 is based on a non-parametric approach; alternatively a parametric method, based on the same assumption that ancient flood events follow the same statistical behavior of those systematically recorded, could have been considered. See Stedinger and Cohn (1986) and Francés

It is indeed true that a non-parametric approach could have been considered. However, in this paper we had the preference to create a continuous data set instead. This is because, since recently, the Dutch water policy uses a new method in which a continuous data set of 50,000 years based on resampled measured weather conditions (e.g. rainfall, temperature, evapotranspiration) is used to predict flood frequency relations (Hegnauer et al., 2014, and also described in Chbab (2006)). We wanted to use the method of Hegnauer et al. (2014) of creating a continuous data set to test whether it also works with resampling measured discharges. This makes the use of HBV and hydraulic modelling to translate the weather data into maximum discharges redundant, as was done by Hegnauer et al. (2014).

Furthermore, we wanted to create a continuous data set since the computation of the confidence intervals of a flood frequency relation remains unchanged compared to the analysis of just measured annual maximum discharges, making the comparison between the two more reasonable an better understandable for decision makers. This argument has been added in the introduction on page 2 lines 25-28. For future work, it is interesting to study how confidence intervals deviate between the proposed methodology and a method based on a parametric approach. However, our results are in line with the findings of Francés (1998), who also showed that the uncertainty intervals of FFAs reduces if historical information is included in the analysis.

8. L. 2-7, p. 9. The Authors states that "the available goodness-of-fit tests for selecting an appropriate distribution function are often inconclusive. Those tests are more appropriate for the central part of the distribution than for the tail (Chbab et al., 2006), where we are interested in since the tail determines the investments required for future flood protection measures." I agree with the Authors

that goodness-of-fit tests might be inconclusive, as discussed deeply in Serinaldi et al. (2018); on the other hand they provide a first indication on which models, among several competing ones, could be excluded due to the poor performance (see, e.g., Laio, 2004). In such a sense, I suggest the Authors at least to rephrase the sentence, also because there are different goodness-of-fit test which focus on the statistical behavior of the tails, such as the Anderson-Darling test and the Modified Anderson-darling test (Laio, 2004).

We agree with you that there are various goodness-of-fit tests, all with their own properties. The sentence has been rewritten with in green the new text:

"A probability distribution function is used to fit the annual maximum discharges to its probability of occurrence. Many types of distribution functions and goodness-of-fit tests exist, all with their own properties and drawbacks. However, the available goodness-of-fit tests for selecting an appropriate distribution function are often inconclusive. This is mainly because each test is more appropriate for a specific part of the distribution, while we are interested in the overall fit of the distribution. This is because the safety standards expressed in probability of flooding along the Dutch dikes vary from $10^{-2}$ to $10^{-5}$." Please see page 11 lines 4-9.

9. Following the argument of previous comment, I do not believe that restricting the analysis to a single probability distribution model (although it is the Generalized Extreme Value distribution commonly used in literature to perform an FFA) is a good choice. Since the interest is in evaluating how the confidence bounds of extreme quantile estimates reduce when adding the historical information (l. 18- 21 p. 9), it should be considered that confidence bounds depend not only on the length and information content of the dataset but also on the probability model itself. Hence, results could be different if a different model is taken into account.

You are indeed correct that the uncertainty interval also highly depends on the fitted distribution itself. Although not shown, we performed the analysis with other distributions as well (e.g. Weibull and Gumbel) and the general conclusion of 'reduction of the confidence bounds as a result of extending the data set of measured discharges' also holds for these distributions. For the GEV distribution we found a reduction of 73% as a result of extending the data set of annual measured discharges with historic events, with the Gumbel distribution a reduction of 60% and with the Weibull distribution a reduction of 76%.

We have added a section to the discussion of the revised manuscript in which it is stated that also for other distribution functions a reduction of the confidence interval was found (page 18 lines 1-7. However, we will not show the in-depth results of different distribution types, because we think this is distracting the reader from the analysis performed and corresponding main findings. Furthermore, the GEV distribution has been shown to fit the data of the Rhine river well and therefore this distribution was preferred above other distributions. Finally, we would like to highlight that many closely-related studies also only focused on the use of a single distribution (e.g. Francés (1998)).

10. L. 10-12 p. 9. Do you the Authors mean that they assume an upper bounded distribution? This issue should be clarified.

Yes, we indeed assume an upper bounded distribution. The GEV distribution has an upper bound as a result of the shape parameter which both influences the skewness and kurtosis of the distribution. We use a bounded distribution since the maximum discharge that is capable of entering the

Netherlands at Lobith is limited to a physical maximum value. The crest levels of the dikes along the Lower Rhine are not infinitely high. The height of the dikes influences the discharge capacity of the Lower Rhine and hence the discharge that can flow towards Lobith. This explanation has been added to the revised manuscript (page 11 lines 15-18 and page 12 lines 1-2) such that it becomes clear why we use an upper bounded distribution. The effect of wave overtopping and dike breaches on the discharges at Lobith are explained in high detail by Bomers et al. (2019).

11. Figure 5 is unnecessary, It could be removed.

Figure 5 has been removed from the revised manuscript.

12. Figure 6. The largest extreme events are not included in the uncertainty bounds. The corresponding sample bounds could be included as well to text the model performance (see comment 9).

Also the largest extreme events are included in the uncertainty bounds (see table 1). However, since the upper bound of the measured data set has a value of 29,631 m3/s (table 1) this line was not entirely drawn. Since it leads to confusion, the entire line has been plotted in the revised manuscript. See the figure below.

[Figure]

13. Section 5.2. I am not sure I fully understood the rationale and the approach behind the analysis performed here. The historical events are some of the highest events observed in the whole observation period. If a sample is reconstructed by simply resampling the events observed in 1901-2018 (without including the largest historical events but with the same length of that used in previous sections), the largest events might only be those observed in the more recent period; as a

consequence, the fitted model is expected to be characterized by, e.g., a smaller variance, which implies narrower uncertainty bounds. I do not see this behavior in figure 7 (upper panel). What I see in figure 7 is that the fitted model in the two cases is almost the same, while the uncertainty bounds are significantly different. I can explain this only if the reconstructed samples have a very different length. Please provide a deeper explanation.

You are indeed correct that we simply resample the events observed in 1901-2018 without including the largest historical events but with the same length. This corresponds with the line 'Q$_{Bootstrap}$'. This data set has a length equal to the 1317-2018 period. If we compare the line with the 1317-2018 data set, we indeed see that the uncertainty interval of the Q$_{Bootstrap}$ is still larger even though the length of the two data sets are the same. It must be noted that not only the length influences the uncertainty interval, but also the discharges within the data set and resulting variance.

For the Q$_{Bootstrap}$ data set, the entire measured data set (1901-2018) is used for resampling. The created continuous series (5,000 in total for convergence reasons) has an average variance of 4,19 x 10$^6$ m$^3$/s. For the 1317-2018 data set, only the discharges below a certain threshold in the measured time period (1772-2018) are used for resampling. In this study, the perception threshold was chosen to be equal to the lowest flood event in the historical time period having a discharge of between 6,928-10,724 m$^3$/s. Hence, the missing years in the historical time period are filled with relatively low discharges, but some of the largest events in the historical time period are larger than ever measured. The total variance of the data set decreases (3.35 x 10$^6$ m$^3$/s) as a result of the lower discharges to create the continuous data set. As a result of the lower variance, also the uncertainty bounds are smaller compared to the Q$_{Bootrstrap}$ data set. This explanation has been added to the revised manuscript. Please see page 17 lines 3-10.

14. L. 20-22 p. 14. It is not clear how the extended data set with normalized reconstructed discharges can capture the long-term climatic variability (see also previous comments).

The historic flood events are only normalized for changes in the river system. As a result, the normalized discharges still capture the climatic conditions in the historical time period. Although the missing years within the historical time period are filled with the measured data set 1772-2018, the most extreme events still capture the climatic variability in the period ~1300-2018. This has been added on page 17 lines 13-14.

15. L. 35, p. 14. Isn't it the 1374 event?

The 1374 flood event is indeed the largest observed discharge (at Cologne) of the last 1,000 years. However, in this analysis we consider the largest measured discharge (measurements have been performed since 1901), which correspond with the 1926 flood event. We now refer to Figure 1 to make this clear, see page 14 line 20.

16. Fig. 8. Adding one event equal to the largest one over a record is expected to affect somewhat the estimated model if the record is 100 years while non changes in the model are expected if the record is about 700 years. Hence, which is the lesson learned from this analysis?

The lesson learned is that flood safety assessments become more robust if the data set of annual maximum discharges is extended. After the 1993 and 1995 flood events of the Rhine river, the flood

frequency relation altered significantly resulting in an increase of the design discharge at Lobith of 1,000 m$^3$/s. Such an increase in the design discharge requires huge investments to cope with the new flood safety standards which were set after the 1993 and 1995 floods. Such an increase was not found if a longer time series was included in the analysis. Looking at the results, decision makers might have taken a different decision. This has been added on page 14 line 26-27.

17. Within the Conclusion Section a detailed list of the limitations of the approach proposed here should be provided.

We now discuss the most important drawbacks and assumptions of the proposed method. Please see the discussion section. We focus on:

- The added value of normalized historic flood events.
- Resampling the systematic data set
- The use of a single distribution function and goodness-of-fit test
- Length of the extended data set and chosen perception threshold
- Comparison with Bayesian statistics.

"Meurs (2006) has reconstructed maximum discharges during historic flood events near the city of Cologne (Germany). The oldest event dates back to 1342. The used method is described in detail by Herget and Meurs (2010), in which the 1374 flood event was used as a case study. Historic documents providing information about the maximum water level during the flood event were combined with the reconstruction of the river cross section at that same time. Herget and Meurs (2010) calculated mean flow velocities near the city of Cologne at the time of the historic flood events with the use of the empirical Manning's equation:

$$Q_p = A_p R_p^{2/3} S^{1/2} n^{-1}$$

where $Q_p$ represents the peak discharge, $A_p$ the cross-sectional area during the highest flood level, $R_p$ the hydraulic radius during the highest flood level, S the slope and n the Manning's roughness coefficient.

However, the highest flood level as well as Manning's roughness coefficient are uncertain. The range of maximum water levels was based on historical sources, whereas the range of Manning's roughness coefficients were based on the tables of Chow (1959). With this information, Herget and Meurs (2010) were able to calculate maximum discharges of the specific historic flood events and associated uncertainty range (Fig. 3)."

The reconstructed historic discharges and their uncertainties were used as input data of the 1D-2D coupled model to compute resulting discharges at Lobith. This is a valid method since there is a strong correlation between the discharge at Cologne and Lobith for in channel flow conditions, even though Cologne is located roughly 160 km upstream of Lobith since they are located in the same fluvial trunk valley and only have minor tributaries (Sieg, Ruhr and Lippe) joining in between (Toonen,

2015). This has been added in the revised manuscript (page 6 lines 10-13) to clarify the applicability of using historical discharge reconstructions at Cologne to determine corresponding present-day maximum discharges at Lobith.

With the 1D-2D coupled model, a Monte Carlo analysis was performed for each historic flood event in which the following parameters were considered to be random: maximum upstream discharge (based on the uncertainty range of each historic flood event as reconstructed by Herget and Meurs (2010)), dike breach thresholds, dike breach formation time and final breach width. The method of this analysis is discussed in detail by Bomers et al. (2019). As a result of the uncertain upstream discharge and breach characteristics, also the discharge at Lobith for each historic event is uncertain. Therefore, many model runs are performed for each event until convergence in model results is reached. Hence, the expected discharge at Lobith and expected 95% confidence intervals were computed (Fig. 3). The hydraulic modelling approach to normalize the historic flood events is now explained in more detail in the revised manuscript. Please see section 2.3.2 and Figure 3.

The authors propose a bootstrap sampling method to fill the gaps between the historic floods with annual flood peaks from the systematic data set, that have an expected value lower than the sampled perception threshold which is set as the smallest flood among the historic peaks. This is approach seems to be critical as it does not add any information to the statistical analysis. The todays conditions are modified by the first extension to the part of the series until 1772. With the sampling the authors accept that the flood series consist of independent and identically distributed random variables, which is not certain.

Indeed, we assume independent and identically distributed random variables. The authors are aware of this assumption. However, please note that to perform a flood frequency analysis we always have to assume that the discharge observations are independent and stationary (Khaliq et al., 2006). Although the assumption is highly uncertain, it must be noted that up till now no consistent large-scale climate change signal in observed flood magnitudes has been identified (Blöschl et al., 2017) justifying the assumption of independent and identically distributed random variables. We have added this in the discussion section. See page 17 lines 21-24.

By definition bootstrapping is any test or metric that relies on random sampling with replacement. Here the wording "resampling of the non-systematic time series below the perception threshold" would be more appropriated. This has been done 5000 times and also the historical floods are varied within their 95% confidence intervals (however these were estimated!). The systematic series were not changed.

Maybe this was not fully clear to the reviewer, but also the systematic data set was changed. For each year within the historical period of which no data is available, an annual maximum discharge of the systematic data set below the perception threshold was randomly drawn (See step 5 in Fig. 4). This corresponds with the bootstrap method. As a result, each created continuous data set is different. We have now explained step 5 in more detail on page 10 lines 25-27

Furthermore, the values within the systematic data set were varied within their 95% confidence intervals. The study described the uncertainties of the systematic data set, which vary for different time periods as a result of different measurement methods used. Please see Fig. 1 in the manuscript

and Section 2.1. Table 1 has been added to the manuscript in which all types of uncertainties are described for the various data sets used to extend the data set of maximum discharges.

The GEV was estimated for each of these samples, the distributions were averaged (!) and their 95% percent confidence bounds were estimated. Table 1 specifies these 95% bounds with the 2-sigma-reach, this would be only justified if the quantiles would be normal distributed. I suppose that this is not the case.

Indeed, the confidence bounds of the discharges are not normally distributed. The caption of the table stating 2-sigma is not correct. If you look at the numbers of the uncertainty bounds you can already see that the confidence bounds are not normal distributed since the upper bound is much further away from the average value than the lower bound, specifically for a return period of 100,000 years. The caption has been changed accordingly in the revised manuscript (see table 2).

In total the value of this resampling study stays unclear for me as it does not extend the information content. The information, derived from the systematic series are used in a simulation study, but the basic assumption that the floods between 1772 and 1900 are reconstructed correctly adds uncertainty to it.

We indeed assume that the 1772-1900 flood were reconstructed correctly, but not without uncertainty (Figure 1). We have included this uncertainty in the analysis. The 95% bounds of the 1772-1900 data set are determined by Toonen (2015) and explained in more detail on page 4 lines 9-12 and page 5 lines 1-2. He found an uncertainty interval of approximately 12%. This has been added explicitly to the revised manuscript to avoid further misunderstanding. Furthermore, we have added a table describing the uncertainties of the various data sets used to extend the systematic data set (see table 1).

There are at least two other options to consider historic floods in statistics:

REIS D. S., JR.; STEDINGER J. R. (2005): Bayesian MCMC flood frequency analysis with historical information. In: Journal of Hydrology, 313, pp. 97–116 (cited by the authors)

Wang, Q. J. (1990): Unbiased estimation of probability weighted moments and partial probability weighted moments from systematic and historical flood information and their application to estimating the GEV distribution. In: Journal of Hydrology 120 (1-4), S. 115–124

Both methods combine the information from the systematic data with historic floods without assumption that these observations are representative for the historic series. In both methods, it is assumed that the historic floods are representative for today's conditions. These events are used to improve the estimation of the upper tail only. The systematic part of the series stays untouched. In this way the uncertainty of assumptions of a large part of the time series is avoided. The statement of the authors: "Most studies found that the confidence intervals of design discharges were reduced significantly by extending the systematic data set with historic events." does not mean that an artificially extended systematic dataset would be beneficial if it was expanded with uncertain assumptions about past flood conditions and their adaptation to the current situation.

We agree with you that we add uncertainty to the data set by adding historical flood events to the measured data set and by using a resampling method to create a continuous data set. However, it must be noted that many of the uncertainties of the historic flood events are included in the analysis, as well as the uncertainty of the systematic data set (1772-2018). An overview of the uncertainties considered is now given in table 1. The 95% confidence intervals of the flood frequency relations are hence based on these uncertainties.

It is true that our method influences the flood frequency curve in the domain of the systematic data set (discharges with high probability of occurrence). However, as far as we know this is always the case if the parameters of the (GEV) distribution are recomputed as a result of new data availability. If we have a look at the figure below, we find that the design discharge with a return period of 100 years decreases from ~12,080 m³/s to ~11,630 m³/s by extending the systematic 1901-2018 data set towards 1317 using the bootstrap method. This decrease in design discharge corresponds with a change of 3.7% indicating that resampling the systematic data set of the historical time period only has a little effect on the shape of the flood frequency curve corresponding with high probability of occurrence. This justifies the use of the bootstrap method. Furthermore, we would like to highlight that we are typically interested in correct prediction of the tail, rather than the discharges with large probability of occurrence, since the tail (high return periods) is of high importance to design flood protection measures. We have added this information in the discussion section on page 19 lines 4-12.

[Figure]

My summary: The manuscript has some weakness with regard to uncertainty assessments (confidence intervals) where the methodology is not sufficient described. The assumption of a symmetrical interval seems to be arbitrarily. Nevertheless the topic is interesting, the manuscript should be consider the existing state of the art in this field and compare its results with well-established existing methods. I suggest to reject the manuscript for major revisions.

We agree with the reviewer that we did not provide enough details about the considered uncertainties of the various data sets used. We have provided a detail explanation of the computed 95% confidence intervals of the following data sets in the revised manuscript (see section 2):

- Reconstructed historic flood events at Cologne by Meurs (2006)
- Corresponding historic discharges at Lobith using a hydraulic model
- Reconstructed discharges for the period 1772-1900 by Toonen (2015)

- Measured discharges for the period 1901-2018

Furthermore, we have added more information about why we propose this method instead of a Bayesian method in the introduction on page 2 lines 4-23. We would like to highlight that our method is systematic. We can extend our data set with historical data and keep the method of a flood frequency analysis the same. In this way, we can make a clear comparison on the effect of extending the data set with multiple other sets on the confidence bounds of flood frequency analysis (page 2 lines 24-30).

Although the maximum likelihood method only gives a point estimate of the (GEV) parameters, as sample size increases, maximum likelihood estimators become unbiased minimum variance estimators with approximate normal distributions. This is used to compute confidence bounds for the GEV parameter estimates. We would like to highlight that, although the Bayesian method is capable of predicting parameter uncertainty without the assumption of being normally distributed, the results are influenced by the prior. The influence of the prior, which has to be defined by the modeler, on the posterior distribution of the parameters and hence on the uncertainty of flood frequency relations can even be larger than the influence of discharge measurement errors, as was found by Neppel et al. (2010). The disadvantage is thus that we have to choose the prior in the Bayesian method correctly such that the tail will be correctly predicted. However, we do not have any measurements in, or near to, the tail and consequently it is reasonable to estimate the prior by fitting the original data with the use of e.g. the Maximum Likelihood method. In this way, the benefits of the Bayesian method compared to a traditional flood frequency analysis are at least questionable. We have added a this to the discussion in section 6.5 and the introduction on page 2 lines 14-20.

We are aware that there is a strong debate between the 'Bayesians' and the 'Frequentist' in literature and discussion forums. With this paper, we do not want to get into this discussion. Rather, we wanted to show a novel and systematic approach which is easy to understand for practitioners to include historic flood information into flood safety assessments. The general methodology of a flood frequency analysis remains in this proposed bootstrap methodology, only the data set of measured discharges is extended. As a result, this method is close to current practice of water managers. We have added the reasons why we set up a bootstrap method in the introduction of the revised manuscript and compared the methodology with the Bayesian statistics briefly in the discussion in section 6.5.

**Comments by reviewer 3**

**[General reply from the authors]**

We would like to thank the anonymous reviewer for taking the time to review our manuscript. We highly appreciate the suggestions and comments, which are helpful in improving the manuscript. Below we have replied to the various comments made by the reviewer.

**[Replies to reviewer comments]**

In this paper, the authors present a method/case study to reconstruct a continuous times series of annual maximum discharges in order to estimate return times for flood discharges for the Rhine at Lobith. The study uses modern data from 1901 onwards, discharges reconstructed from water level measurements back to 1772 and information from historical flood events back to the 1300. Extending a time series with this information leads to a reduction of uncertainty and to more stable return times. The paper is well structured and written, and the topic is of relevance for flood risk estimation.

However, there general problem I have with this manuscript is that the authors refer to and use data from many other studies, especially the one from Toonen (2015). It is difficult to follow the article for reader if one is not familiar with these studies because it requires reading many secondary sources to gain insight on how all the different data(-sets) were collected and obtained, e.g. how was the regression analysis by Toonen (2015) performed, how were the historical floods in Cologne by Herget and Meurs (2010) reconstructed, etc. This paper includes a lot of different data sets (systematic, historical, plus various bootstrapped time series), it would be beneficial for readers to include a table with a short description and overview of the properties of these data sets and to name them consistently throughout the paper.

Thank you for this remark, we fully agree with you. In the revised manuscript we have provided more knowledge about how the discharges at Lobith were reconstructed by Toonen (2015) (page 4 lines 9-12 and page 5 lines 1-2) as well as the reconstructions at Cologne performed by Herget and Meurs (2010) (page 6 lines 1-6).  Furthermore, the following table has been added to the revised manuscript as also suggested by Elena Volpi (first reviewer):

**Table 1.** Uncertainties and properties of the various data sets used. The 1342-1772 data set represents the historical discharges, whereas the data sets in the period 1772-2018 are referred to as the systematic data set

| Time period | Data source | Property | Cause uncertainty | Location |
|---|---|---|---|---|
| 1342-1772 | Meurs (2006) | 12 single events | Reconstruction uncertain caused by main channel bathymetry, bed friction and maximum occurred water levels | Cologne |
| 1772-1865 | Toonen (2015) | Continuous data set | Reconstruction uncertainty based on measured water levels of surrounding sites | Emmerich, Pannerden and Nijmegen |
| 1866-1900 | Toonen (2015) | Continuous data set | Uncertainty caused by translation measured water levels into discharges | Lobith |
| 1901-1950 | Tijssen (2009) | Continuous data set | Uncertainty caused by extrapolation techniques to translate measured velocities at the water surface into discharges | Lobith |
| 1951-2000 | Tijssen (2009) | Continuous data set | Uncertainty caused by translation velocity-depth profiles into discharges | Lobith |
| 2001-2008 | Tijssen (2009) | Continuous data set | Measurement errors | Lobith |
| 2009-2018 | Measured water levels available at https://waterinfo.rws.nl | Continuous data set | Measurement errors | Lobith |

The term "normalize" is used in different contexts (e.g. for historical floods, for the 1900-2008 data set, for the data set of Toonen (2015) which is not normalized but used as normalized data). I find this confusing since it does not become clear what is actually meant by this and what has been done to "normalize" each of these data sets. A more thorough explanation on this matter would be useful.

With the term 'normalize' we mean that we translate the historic flood events (water levels, discharges) to present-day discharges at Lobith as a result of changes in the river system and hinterland. Please see also page 2 lines 34-35 where an explanation of the term is given. In the revised manuscript we will explain in more detail how the normalization was done for the various data sets used in this manuscript. The following text has been added with in green te new text:

Regarding the 1901-2008 data set (page 3 lines 16-24):

"Daily discharge observations at Lobith have been performed since 1901 and are available at https://waterinfo.rws.nl. From this data set, the annual maximum discharges are selected in which the hydrologic time period, starting at the 1st of October and ending at the 30th of September, is used. Since changes to the river system have been made the last century, Tijssen (2009) has normalized the measured data set from 1901-2008 to the conditions of the year 2004. In the 20th century, canalization projects were executed along the Upper Rhine (Germany) which were finalized in 1977 (RIZA, 2003). After that, retention measures were executed in the trajectory Andernach-Lobith. Firstly, the 1901-1977 data set has been normalized with the use of a regression function describing the influence of the canalization projects on the maximum discharges. Then, again a regression function was used to normalize the 1901-2008 data set for the retention measures (RIZA, 2003). This results in a normalized 1901-2008 data set for the year 2004."

Regarding the Toonen (2015) data set (page 5 lines 3-5):

"The reconstructed discharges in the period 1772-1900 represent the computed maximum discharges at the time of occurrence and have not been normalized for changes in the river system and thus they represent the actual occurred annual maximum discharges."

Regarding the Herget and Meurs (2010) data set (page 6 lines 17-19):

"In this study, the 1D-2D coupled modelling approach as described by Bomers et al. (2019) is used to normalize the data set of Meurs (2006). This normalization is performed by routing the reconstructed historical discharges at Cologne over modern topography to estimate the maximum discharges at Lobith in present times."

In section 2.2 the authors describe the Toonen (2015) data set which uses a linear regression to compute water levels at Lobith. This method leads to a reduced variance of this data set (c.f. table 1). How would this affect the bootstrapping later on, if samples from the so called "systematic time period" with different variances (1772-1900, 1901- 2018) are drawn?

The Toonen (2015) data set indeed has a lower variance compared to the 1901-2018 data set. To identify the effect of using both data sets for resampling purposes, we have performed an additional FFA in which now only the 1901-2018 data set is used for resampling. The results are presented in the figure below in which the purple line indicates the situation in which only the 1901-2018 data set is used for resampling and the blue line represents the reference situation in which the 1772-2018 is used for resampling.

We can see that using the 1772-2018 results in a reduction of the confidence intervals caused by the lower variance in the 1772-1900 data set. This reduction is at maximum 12% for the return period of 100,000 years. This finding has been added to the discussion on page 17 lines 27-34.

However, do note that the lower variance in the 1772-1900 period compared to the 1901-2018 period is most probably a result of natural variability in climate. It is this variability that we want to include in the analysis since also climate variability will exist in the future. If the lower variance was caused by e.g. the removal of a dam construction upstream, it would be reasonable to solely use the 1901-2018 data set for resampling purposes.

[Figure]

From my point of view, the section 2.3.2 presenting the normalization of historical flood events leaves some open questions which need to be addressed. Using a coupled 1D/2D model to route the discharges from Cologne to Lobith seems a reasonable approach given the circumstances of the data, but the dike breach model and the underlying assumptions need more explanation. Is it valid to assume dike breach parameters from today's river geometry for historical times? Is there any historical evidence that there were dike breaches in the past, especially the 1374 event? Especially the reduction of the 1374 flood peak from Cologne to Lobith needs some sound justification/explanation. Why is this reduction only occurring for this specific event? Were there also dike breeches for the other historical events?

Please note that de 1D-2D coupled model is only based on the current geometry and current dike strengths. This is because only then normalization can be performed. So, whether dike breaches occurred during the historical flood events between Andernach and Lobith may be interesting from historical point of view (e.g. a reconstruction of this flood in historical times), but is not directly relevant for this study, as we are interested what will happen nowadays. Therefore, we use so-called fragility curves showing at which water level the dikes in the studied area will start to breach. We now provide more insights in the 1D-2D coupled modelling approach in section 2.3.2 and particularly about the dike breach parameters (please also see figure 3).

Concerning the 1374 flood event, this event results in a large reduction of the maximum discharge because major overflow and dike breaches occur in present times. Since the 1374 flood event was much larger than the current discharge capacity of the Lower Rhine, the maximum discharge at Lobith decreases. On the other hand, the remaining flood events were below this discharge and hence only a slight reduction in discharges were found for some of the events as a result of dike breaches whereas overflow did not occur. Other events slightly increased as a result of the inflow of the tributaries Sieg, Ruhr and Lippe rivers along the Lower Rhine. This explains why the 1374 flood event is much lower at Lobith compared to the discharge at Andernach, while the discharges of the

remaining flood events are more or less the same at these two locations. This information has been added to the revised manuscript on page 8 lines 21-22 and page 9 lines 1-7.

What exactly is meant by "the upstream discharge shape is varied" (p.6, line 12)? There is a lot of uncertainty in this, which somehow contradicts the aim of the paper to reduce uncertainty.

Of the historic flood events at Cologne, only the peak value was known. The corresponding shape of the discharge wave was unknown. However, this shape may affect the maximum discharge at Lobith. Therefore, we want to include this uncertainty in the analysis. Although it is indeed true that we wanted to reduce uncertainty in flood frequency relations, it does not mean that we want to ignore known uncertainties in the reconstructions.

We used a data set of 250 potential discharge shapes that can occur under current climate conditions (Hegnauer et al., 2014). See the figure below for an example of three potential discharge shapes: e.g. a broad peak, a small peak or a discharge wave with two peaks. For each run in the Monte Carlo analysis, we randomly sampled a shape and scaled this shape to the maximum value of the flood event. This represents the upstream boundary condition of the model run. We now provide more information about the upstream discharge wave shapes on page 7 lines 8-14. Please also see figure 3.

[Figure]

Furthermore, it would be interesting to know if any of 12 historical flood events where winter events, where ice draft/ice jams could/did play a role.

All flood events were winter events, except for the flood event in 1342 that took place in July. However, the flood events caused by ice jams were excluded from the analysis by Herget and Meurs (2010) because of the different hydraulic conditions. All flood events considered are thus caused by high rainfall intensities. This has been added on page 5 lines 12-13.

Furthermore, assuming a normal distribution of uncertainties is valid for discharge measurements, but is this also the case for the estimation of historical extreme floods? Or is any discharge values in the uncertainty range equally possible? The reconstruction of the events in Cologne is based on the Manning equation and the uncertainty range results from different roughness coefficients. But do all of these follow a normal distribution?

Herget and Meurs (2010) only provided the maximum, minimum and mean value of the roughness coefficients. They did not provide any insights in the distribution of this uncertainty. We assumed

that they were normal distributed since it is likely that the mean value has a higher probability of occurrence than the boundaries of the considered range. This assumption results in a normal distribution of the maximum discharge at Andernach and consequently to a normal distribution of the maximum discharge at Lobith.

However, we performed the resampling bootstrap method in a different way. During the resampling we assumed uniformly distributed uncertainties and we re-performed the analysis with normally distributed ones. The difference between the two is given in the figure below. We find that assuming normally distributed uncertainties results in slightly smaller uncertainty bounds which can be explained by the lower variance. However, this effect is only very little justifying the assumption of normally distributed uncertainties. This has been added on page 7 lines 3-7.

[Figure]

Section 3: The bootstrap method to create continuous times series is a reasonable approach, however it would also be possible, to use the maximum likelihood method and incorporate the uncertainty range of the historical discharges as well as the discharges lower the perception threshold in the parameter estimation. From my point of view this approach is straight forward and

should yield similar results. Could the authors explain/discuss the benefit of the bootstrapping approach?

We have created a continuous data set by incorporating the uncertainty range of the historical discharges as well as the discharges lower than the perception threshold. Next, we have used the maximum likelihood method to fit each continuous data set (we have 5,000 in total) to a GEV distribution (please see figure 5 and the explanation of the bootstrap method in section 3). We do not understand the difference between our method and the method suggested by the reviewer. If our method was not fully understood by reading the manuscript, we will make this clearer in the revised manuscript.

In Section 4, the authors state that there are many distributions and fitting methods for flood frequency analysis and that the only use the GEV with maximum likelihood method. It seems justified, that only one combination is used to quantify the reduction of the uncertainty, but in practice there are many different distributions and parameter estimation methods - which again cause higher uncertainties in the estimation of return times, especially for the upper tail extremes. The authors should include a comment and if possible a quantification of this effect on this in the discussion.

You are indeed correct that the use of various kinds of distributions and parameter estimation methods influence the uncertainty in the flood frequency relations. We have performed the analysis with the Gumbel and Weibull distribution as well and these results are now shown in the discussion. We will also highlight that using the combination of multiple distributions in the analysis increases the uncertainties in the estimation of return periods. We have added this in the discussion of the revised manuscript in section 6.3.

In Section 5.2., the authors argue that the reconstruction of historical flood events is complicated and time consuming and that this can be overcome by bootstrapping. However, the information from rare and large historical flood events is still required as is stated at the end of the section. This sounds like an inconsistency in the line of argumentation. Furthermore, this whole section is somewhere between results and discussion. I suggest that the authors try to separate more clearly between results and discussion.

We indeed argue that reconstructing historic flood events is time consuming. Therefore, we studied whether it is also possible to only use the 1901-2018 measured data set in a bootstrap approach. However, we find that the uncertainty interval of this FF curve is larger than for the FF curve in which the normalized historic flood events are considered. We thus show that, although it is time consuming to normalize the historic flood events, it is worth the effort since it reduces uncertainties in FF relations. Since this was not fully clear, we have rewritten the paragraph in the revised manuscript. Furthermore, we have rewritten the paragraph in a more discussion style and replace this section towards the discussion section. Please see section 6.1.

In the discussion, the effect of a hypothetical future extreme flood on the robustness of return times is addressed, which is somehow obvious from my point of view. This aspect does not add much value to the paper and can either be omitted or be moved to the results section.

We have moved this section towards the results. We believe that it shows the robustness of the method since using an extended data set in flood management avoids that a flood frequency curve changes after the occurrence of a future flood event. As a results, the FFA does not have to be performed again, while this is necessary if only the data set of measured discharges is used. Therefore, decision makers might have taken another decision. Please see page 14 lines 26-27.

Some specific comments:

Page 3, line 3f.: Why are uncertainties not symmetrical due to missing continuous data? Don't these result from the non-linearity of the rating curves?

The sentence about the symmetrical uncertainties stated in the introduction was not fully correct. Indeed, uncertainties are in general not symmetrical for flood frequency relations. This is indeed the result of the non-linearity of the rating curves. However, the introduction have been revised significantly and as a result the sentence related to the symmetrical uncertainties have been removed.

Page 4, line 7f.: ACDP-measurements are in general not free of uncertainties, this assumption is not correct.

Indeed, the ACDP-measurements are in general not free of uncertainties. Since we had no reference regarding this uncertainty, we used the uncertainties as suggested by Toonen (2015). He mentioned that only the discharges slightly exceeding the bank-full discharge have an uncertainty range of 5%. In the revised manuscript we now include this uncertainty for all ACDP-measurements (see page 3 lines 31-32 and table 1). However, since all annual maximum discharges in the period 2000-2018 where between 4,000 and 8,000 $m^3$/s, the 5% uncertainty was already included in the analysis and hence the results will not change.

Page 11, line 2: Where does this confidence interval of 7400m3 /s come from?

This value represents the reduction in the confidence interval if the 1901 data set is extended towards 1317 for the discharges corresponding with a return period of 1,250 years. We have rewritten the text such that this becomes clearer. Please see page 12 line 22 and page 13 lines 1-2.

Page 15, line 1: Same as above, modern discharge measurements are not free of measurement errors!

Please see above and page 3 lines 31-32 and table 1.

Page 15, line 5f.: See above, this is not a novel results and can more or less be expected. Furthermore, the statement that "flood managers can be less nervous" sounds awkward and is not really correct, since the uncertainty caused by different distributions/parameter estimation methods is not addressed.

This section has been moved to the results. It is indeed true that we did not include the uncertainty caused by different distributions and parameter estimation methods. We have removed the statement from the manuscript and added in the discussion the effects of using a combination of different distributions on the uncertainty intervals. Please see section 6.3.

Figure 2: Should be replaced by a "conventional" map, including national boundaries, a scale bar etc. Readers from outside of Europe might not be familiar with this region.

The figure has been replaced by the following figure such that now the national boundaries, scale bar, north arrow, names and model boundary are given. Please see figure 2

[Figure]

Table 1: The results of Toonen (2015) can be omitted in this table from my point of view.

The results of Toonen (2015) are omitted from the table.

Figure 6 and 7: The colours/line styles of the different curves are difficult to distinguish and should be changed to make these figures better to read.

The colours are adapted as follow:

[Figure]

[Figure]

References: To my knowledge, Meurs 2006 is a diploma thesis, not a PhD thesis.

You are correct, it is indeed a diploma thesis. This has been adapted in the revised manuscript.

REFERENCES:

Hegnauer, M., Beersma, J. J., van den Boogaard, H. F. P., Buishand, T. A., and Passchier, R. (2014). Generator of Rainfall and Discharge Extremes (GRADE) for the Rhine and Meuse basins. Final report of GRADE 2.0. Technical report, Deltares, Delft, The Netherlands

**List of all relevant changes made in de manuscript**

- Introduction has significantly been changed. We now provide more information about the differences between the Bayesian approach and the frequentist statistics. Also the reasons why we set up a bootstrap method are now better described.
- More information about the data sets used is given as well as the uncertainties involved.
- More information about the hydraulic model has been added in which we now describe in more detail how the normalization steps of the historical flood events are performed.
- The discussion section has been altered a lot. More information about the assumptions and drawbacks are given.

[revised manuscript text omitted]

---

## Author Response (AR2)

Editorial Office
Natural Hazards and Earth System Sciences

Enschede, August 12, 2019

Dear Editor,

Thanks a lot for the two reviews. We have read the revised manuscript carefully and corrected language errors and typos.

Thank you for your kind consideration and please feel free to contact me for any question.

On behalf of all authors,
Yours sincerely,

Anouk Bomers
Department of Water Engineering and Management
University of Twente
P.O. Box 217
7522 NB  Enschede - the Netherlands
Tel.: +31 53 489 1062
Email: a.bomers@utwente.nl